# OUT-OF-DISTRIBUTION ROBUST EXPLAINER FOR GRAPH NEURAL NETWORKS

## ABSTRACT

Graph Neural Networks (GNNs) are powerful tools for analyzing graph-structured data; however, their interpretability remains a challenge, leading to the growing use of eXplainable AI (XAI) methods. Most existing XAI models assume that GNNs are well-trained and that all nodes in the graph share similar data characteristics to those used during GNN training. In real-world applications, new nodes and edges are frequently added to the input graph during testing. This dynamic environment can introduce out-of-distribution (OOD) nodes, potentially undermining the reliability of XAI models. To address this issue, we propose an OOD Robust Explainer (ORExplainer), a post-hoc, instance-level explanation model specifically designed to provide robust and reliable explanations in the presence of OOD nodes, noise, and outliers in graphs. ORExplainer incorporates Energy Scores to capture structural dependencies, allowing for prioritizing in-distribution nodes while reducing the impact of OOD nodes. We conduct experiments with varying types of OOD node inclusion. ORExplainer demonstrates superior robustness of generated explanations across synthetic and real-world datasets. Our code is available at `https://anonymous.4open.science/r/ORExplainer-C52C/`.

## 1 INTRODUCTION

Graph Neural Networks (GNNs) have become essential for modeling graph-structured data in domains such as social networks, biology, and recommendation systems (Feng et al., 2023; Wu et al., 2022). As these models are increasingly used in critical applications (Longa et al., 2024; Yuan et al., 2022), their interpretability has attracted growing attention. To address this need, post-hoc instance-level explanation methods (Ying et al., 2019; Luo et al., 2020) aim to identify subgraphs most influential to predictions, and recent studies (Zhang et al., 2023; Chen et al., 2024) further improve their reliability in high-stakes domains.

While prior efforts have advanced our understanding of GNN decision-making, existing explanation methods often fail to align with real-world scenarios. Most approaches implicitly assume that the explainer model is trained on the same graph as the graph employed to train the underlying GNN model to be explained, an unrealistic setting when applied beyond controlled benchmarks. In practice, real-world graphs can evolve with the addition of new nodes and edges, such as newly published papers in citation networks or newly joined users in social networks. As a result, the graph available to the explainer model may differ from the one originally used to train the GNN. Since papers from entirely new domains or injected unexpected users may constitute out-of-distribution (OOD) instances, it is important to design explainer models that are robust to OOD scenarios.

To systematically analyze explanation robustness, we categorize OOD nodes into three representative types as shown in Figure 1. **Structure-level OOD** occurs when injected nodes alter the graph's connectivity significantly. **Feature-level OOD** arises when new nodes exhibit feature patterns unseen during training. **Unseen-label OOD** refers to nodes belonging to classes absent from the training data. Together, these scenarios represent realistic challenges are useful for evaluating the robustness of node-level explanations. For robust and trustworthy explanations, the explanatory subgraph should primarily rely on in-distribution (ID) evidence, while avoiding OOD instances that the pre-trained GNN cannot reliably process.

In response to these OOD scenarios, we introduce ORExplainer, a robust post-hoc explainer tailored for noisy graphs. ORExplainer extracts compact subgraphs that preserve predictive information

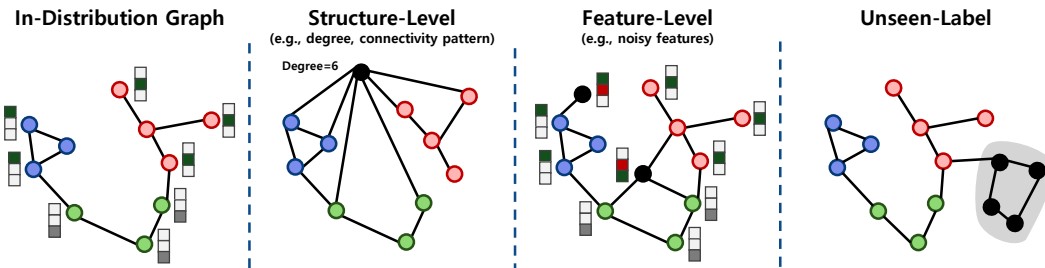

Figure 1: Illustration of different types of out-of-distribution (OOD) nodes in graphs, including structure-level, feature-level, and unseen-label OOD cases.

while reducing the impact of OOD nodes. It quantifies the model's confidence in each node via energy scores (Ranzato et al., 2007; Liu et al., 2020) and introduces weighted energy propagation to capture relational structure, enabling robustness against various type of OOD interference. By focusing on ID nodes, ORExplainer provides reliable explanations even when ID and OOD nodes coexist, distinguishing it from prior explainers. Through extensive experiments on both synthetic benchmarks and real-world datasets under varying OOD conditions, we demonstrate that ORExplainer consistently delivers superior performance, underscoring its effectiveness in practical graph scenarios.

**Contributions.** We summarize our contributions:

- **Robust Explanation in Noisy Environments:** We propose ORExplainer, a post-hoc explainer designed for graphs with OOD or noisy nodes, providing robust explanations by suppressing unreliable information.
- **Analysis of Baseline Vulnerabilities:** We systematically evaluate existing explainers under OOD settings and show that many fail to provide accurate explanations, while ORExplainer addresses these vulnerabilities.
- **Energy-Based OOD Handling:** We introduce *Weighted Energy Propagation (WEP)*, which leverages energy scores to prioritize ID nodes and downweight OOD ones, enhancing robustness and reliability across diverse graph environments.

## 2 RELATED WORK

Explainable AI models in the graph domain focus on identifying substructures that significantly impact outputs from trained models. Primarily, GNNExplainer (Ying et al., 2019), a pioneering study in this field, proposes a mask-based method to find important subgraphs that maximize the mutual information with the predictive output. Furthermore, PGExplainer (Luo et al., 2020) advances this concept by parameterizing explainers in a more generalized setting, approximating multiple important subgraphs for various instances using a single explainer. Additionally, SubgraphX (Yuan et al., 2021) employs Monte Carlo Tree Search to identify important subgraphs with the highest Shapley value.

While various state-of-the-art explanation methods contribute to generating high-quality explanations, another line of research questions have emerged regarding their generalization and robustness. MixupExplainer (Zhang et al., 2023) and ProxyExplainer (Chen et al., 2024) address the issue that explanatory subgraphs often suffer from a distribution shift **relative to** the input graphs, due to differences in size or structural properties. Since a pretrained GNN model cannot properly process such distribution-shifted graphs, the training of the explainer itself becomes problematic. To mitigate this problem, MixupExplainer mixes input graphs with label-irrelevant graphs, whereas ProxyExplainer employs a VGAE (Kipf & Welling, 2016) encoder to enforce in-distribution explanations. In a different approach, HINT-G (Jung et al., 2025) leverages influence functions (Bae et al., 2022; Wu et al., 2023a) to trace how training nodes affect the prediction of a target node, providing explanations grounded in influence rather than subgraph generation.

Despite significant advancements in explainability, many existing methods often overlook the impact of OOD nodes and edges that can arise **within** the input graph. V-InFoR (Wang et al., 2024), unlike prior works, focuses on designing a robust explainer for structurally corrupted graphs. It leverages variational inference to learn robust graph representations in order to address structural

corruption. However, its robustness mainly targets structural OOD and does not extend to other types of corruption, such as feature noise. In addition, since it is originally developed for a graph classification task, its applicability to node-level scenarios such as node injection remains limited.

**Different Setting Compared to Existing Methods:** Most existing explanation methods implicitly assume that the explainer is trained on the same in-distribution graphs as the GNN model. However, real-world graphs are inherently dynamic, continuously evolving through the addition of new nodes and edges. These dynamics naturally introduce OOD components, which existing explainers are not designed to handle. This underscores the necessity of developing explanation methods that are explicitly designed for graphs with newly added OOD nodes or edges. More extensive related work is provided in Appendix A.

## 3 PRELIMINARIES

### 3.1 NOTATION

Let $\mathcal{G} = \{\mathcal{V}, \mathcal{E}\}$ represent a graph, where $\mathcal{V} = \{v_1, v_2, \ldots, v_N\}$ is the set of nodes with $N$ being the number of nodes, and $\mathcal{E} \subseteq \mathcal{V} \times \mathcal{V}$ is the set of edges. Each node $v_i$ has a feature vector $\mathbf{x}_i \in \mathbb{R}^D$ and a label $y_i \in \{1, 2, \ldots, C\}$, where $D$ is the feature dimension and $C$ is the number of classes. The adjacency matrix is defined as $\mathbf{A} = [a_{ij}]_{N \times N}$, with $a_{ij} = 1$ if $(v_i, v_j) \in \mathcal{E}$ and $a_{ij} = 0$ otherwise.

We denote the graph used to train GNN model $f$, as $\mathcal{G}_{\text{GNN}} = \{\mathcal{V}_{\text{GNN}}, \mathcal{E}_{\text{GNN}}\}$. The graph used for explanation, $\mathcal{G}_{\text{explain}} = \{\mathcal{V}_{\text{explain}}, \mathcal{E}_{\text{explain}}\}$, may contain additional OOD nodes and their edges, such that $\mathcal{V}_{\text{GNN}} \subseteq \mathcal{V}_{\text{explain}}, \quad \mathcal{E}_{\text{GNN}} \subseteq \mathcal{E}_{\text{explain}}$.

The model $f$ is a node classifier, where $f(\mathcal{G}, i)$ takes input as a graph $\mathcal{G}$ and a target node index $i$. $f$ consists of two parts: an encoder $f_{\text{enc}}$ and a classifier $f_{\text{cls}}$. The encoder $f_{\text{enc}}(\mathcal{G})$ generates the embedding set $\mathcal{Z} = \{\mathbf{z}_1, \mathbf{z}_2, \ldots, \mathbf{z}_N\}$, where each $\mathbf{z}_i \in \mathbb{R}^H$ denotes the latent representation of node $v_i$, and $H$ represents the dimensionality of the embedding vectors. The classifier $f_{\text{cls}}$ generates a $C$-dimensional vector representing the class probabilities for each node. The predicted class label $\hat{y}_i$ is then determined by applying the $\arg\max$ function to the class probability vector: $\hat{y}_i = \arg\max(f_{\text{cls}}(\mathbf{z}_i))$.

### 3.2 POST-HOC EXPLAINERS FOR NODE CLASSIFICATION

Post-hoc explainers for node classification (Ying et al., 2019; Luo et al., 2020) aim to extract an explanatory subgraph $\mathcal{G}_t^*$ that captures the most informative structure for a target node $v_t$. This is typically formulated by maximizing the mutual information between the model's prediction $\hat{y}_t$ and the candidate explanatory subgraph $\mathcal{G}_t^*$. Since direct optimization is infeasible, explanation methods introduce relaxations and parameterizations to learn edge masks.

GNNExplainer (Ying et al., 2019) directly assigns a soft edge mask $a_{ij}^*$ for each edge $(v_i, v_j)$, optimizing it to minimize the uncertainty of predictions conditioned on the selected subgraph. In contrast, PGExplainer (Luo et al., 2020) adopts a more general and scalable approach: it trains a Multi-Layer Perceptron (MLP) $g(\cdot)$ that receives edge embeddings $[\mathbf{z}_i; \mathbf{z}_j; \mathbf{z}_t]$ as inputs and outputs mask logits $\omega_{ij}$. These logits are reparameterized into probabilistic edge selections, enabling explanation across multiple nodes.

Both methods apply constraints to enhance interpretability and sparsity. Specifically, an $L_1$ penalty on the edge mask encourages compact subgraphs, while entropy regularization pushes mask values towards binary decisions.

### 3.3 ENERGY-BASED OOD SCORING

A softmax classifier can be equivalently expressed as an Energy-Based Model (EBM) (Ranzato et al., 2007; Grathwohl et al., 2020; Du & Mordatch, 2019). For a GNN model $f$, the free energy of a node $v_i$ is defined as

$$E(\mathcal{G}, i; f) = -\log \sum_{c=1}^{C} \exp(f(\mathcal{G}, i)_{[c]}). \tag{1}$$

This formulation allows the energy to be directly computed from model logits without additional training, and it has been widely used as an OOD score. In this setting, ID nodes generally obtain lower energy values, whereas OOD nodes yield higher energy (Liu et al., 2020; Wu et al., 2023b).

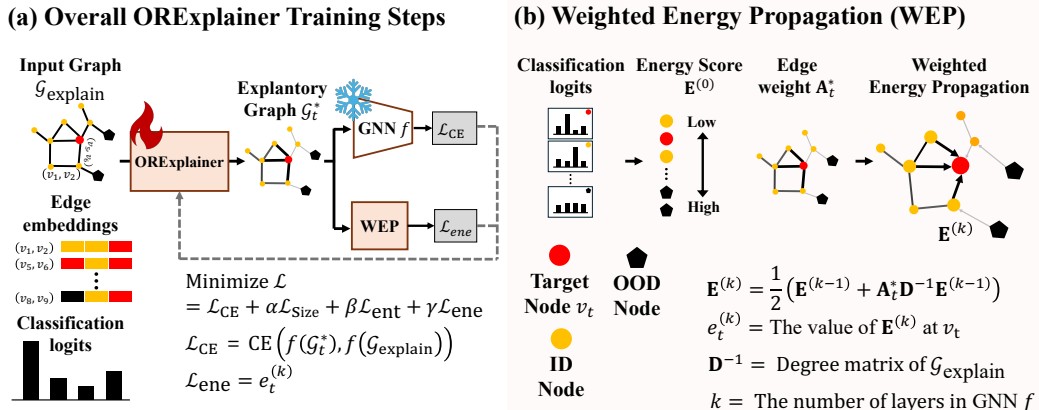

Figure 2: (a) illustrates the overall framework of ORExplainer and (b) details *Weighted Energy Propagation (WEP)* to reduce the effect of OOD nodes (edges).

## 4 OUR PROPOSED METHOD

We propose **O**ut-of-distribution **R**obust **E**xplainer termed as ORExplainer, a post-hoc explanation method for node classification in graphs where both ID and OOD nodes coexist at inference time, though not during training. ORExplainer generates explantions for ID targe nodes whose predictions remain stable under OOD contamination. Under this setting, ORExplainer provides explanations that are faithful to the model's decision while remaining robust to noise and the presence of OOD nodes. An overview of the training framework of ORExplainer is illustrated in Figure 2-(a). We next formalize the problem setting and describe how robust explanations are defined.

### 4.1 ROBUST EXPLANATION FOR NODE CLASSIFICATION

Given a pre-trained GNN $f$ trained on $\mathcal{G}_{\text{GNN}}$ and a target node $v_t$, the goal of an explanation model $g$ is to identify a subgraph that accounts for the prediction of $f$ on $v_t$, where $v_t \in \mathcal{V}_{\text{explain}}$. $v_t$ is an ID node that is correctly classified. This condition suggests that the OOD components have a limited effect on the model's decision-making process. Consequently, incorporating OOD nodes into the explanation subgraph can undermine its faithfulness.

Formally, the robust explainer $g$ can be defined as

$$\mathcal{G}_t^* = g(f, \mathcal{G}_{\text{explain}}, t). \tag{2}$$

$\mathcal{G}_t^*$ denotes the explanation subgraph and $t$ is the index of the target node $v_t$. The key requirement is that $\mathcal{G}_t^*$ should (i) preserve the predictive behavior of $f$ for $v_t$, while (ii) minimizing the impact of OOD nodes that may affect $v_t$.

To instantiate $g$, we adopt a parameterized framework based on an MLP that takes edge embeddings as input. For each candidate edge $(v_i, v_j)$ with respect to a target node $v_t$, the edge embedding is constructed by concatenating the node representations of $v_i$, $v_j$, and $v_t$. Unlike prior methods (Luo et al., 2020; Zhang et al., 2023) that only utilize the final-layer representation of the encoder $f_{\text{enc}}$, our model concatenates intermediate embeddings from all layers of $f_{\text{enc}}$ together with the raw features of the nodes involved. This design provides the explainer with richer multi-scale information, enabling more expressive and reliable explanations. The MLP outputs a scalar logit $\omega_{ij}$ for each edge $(v_i, v_j)$, which is mapped via a sigmoid to a probabilistic edge mask $a_{ij}^* \in [0, 1]$, forming the weighted adjacency matrix $\mathbf{A}_t^* = [a_{ij}^*]$. During training, we employ the Gumbel-softmax reparameterization (Jang et al., 2017) for sampling.

We optimize a cross-entropy loss $\mathcal{L}_{\text{CE}} = \text{CE}\big(f(\mathcal{G}_t^*), f(\mathcal{G}_{\text{explain}})\big)$, ensuring fidelity between the explanatory subgraph and the original graph $\mathcal{G}_{\text{explain}}$. While $\mathcal{L}_{\text{CE}}$ ensures fidelity and interpretability, it does not consider the reliability of explanations under OOD interference. To complement it, we introduce an energy-based scoring mechanism that accounts for OOD nodes.

## 4.2 WEIGHTED ENERGY PROPAGATION

To enhance robustness against various types of OOD interference, we design *Weighted Energy Propagation (WEP)*, which restricts the impact of nodes with unreliable prediction logits. The objective is to construct explanations that emphasize information from ID nodes while suppressing contributions from OOD nodes.

Let $\mathbf{E}^{(0)} = [e_i^{(0)}]_{i=1}^N$ denote the initial energy scores of all nodes in $\mathcal{G}_{\text{explain}}$, where $e_i^{(0)} = E(\mathcal{G}_{\text{explain}}, i; f)$ is obtained from the pre-trained GNN. Energy scores are then propagated through the explanatory subgraph according to

$$\mathbf{E}^{(k)} = \tfrac{1}{2}\left(\mathbf{E}^{(k-1)} + \mathbf{A}_t^* \mathbf{D}^{-1} \mathbf{E}^{(k-1)}\right), \tag{3}$$

where $\mathbf{A}_t^*$ is the weighted adjacency matrix produced by the explainer for target node $v_t$, and $\mathbf{D}^{-1}$ is the inverse degree matrix of $\mathcal{G}_{\text{explain}}$. This formulation ensures that each node retains part of its own energy while also aggregating energy from its neighbors. From the perspective of the target node, connections to low-energy (ID) neighbors reduce its propagated energy, whereas connections to high-energy (OOD) neighbors increase it. By enforcing the target node's propagated energy score to be minimized, the explainer is guided to prioritize information from ID neighbors while suppressing that from OOD neighbors as shown in Figure 2-(b). This is achieved by introducing the robustness term

$$\mathcal{L}_{\text{ene}} = e_t^{(k)}, \tag{4}$$

which penalizes highly propagated energy at the target node $v_t$. Importantly, because the energy score quantifies how confidently the GNN processes each node, this mechanism is not restricted to any single type of OOD (e.g., structural, featural, or unseen-label), but can adapt across diverse scenarios. By explicitly aligning the explanation process with the GNN's own confidence, *WEP* ensures that the resulting subgraph highlights informative ID neighbors while systematically suppressing spurious OOD effects. This robustness term is then incorporated into the overall explainer objective, described in the next section.

## 4.3 EXPLAINER LOSS

The explainer is trained with a composite objective that combines $\mathcal{L}_{\text{CE}}$, with our robustness term from Weighted Energy Propagation. To prevent trivial solutions, we additionally impose an $L_1$ size loss $\mathcal{L}_{\text{size}}$ and an entropy loss $\mathcal{L}_{\text{ent}}$ from Ying et al. (2019); Luo et al. (2020) on the explanation mask $\mathbf{A}_t^*$. The final objective is

$$\mathcal{L} = \mathcal{L}_{\text{CE}} + \alpha \mathcal{L}_{\text{size}} + \beta \mathcal{L}_{\text{ent}} + \gamma \mathcal{L}_{\text{ene}}, \tag{5}$$

where $\alpha, \beta, \gamma$ are hyperparameters controlling the trade-off among size, entropy, and robustness terms.

## 5 THEORETICAL ANALYSIS

We formalize how *Weighted Energy Propagation (WEP)* in Eq. 3 induces a lazy *substochastic* diffusion on the explanatory graph and why minimizing the propagated energy at the target, $\mathcal{L}_{\text{ene}} = e_t^{(k)}$, suppresses OOD influence while preserving faithfulness under the composite loss in Eq. 4. We first establish that the WEP operator $\mathbf{P}_t = \tfrac{1}{2}(\mathbf{I} + \mathbf{A}_t^* \mathbf{D}^{-1})$ is lazy and *column-substochastic*.

**Lemma 5.1** (Column-substochastic laziness)**.** $\mathbf{P}_t$ *satisfies* $\sum_i (\mathbf{P}_t)_{ij} \leq 1$ *for every $j$, with equality iff* $\sum_i (\mathbf{A}_t^*)_{ij} = d_j$, *and* $(\mathbf{P}_t)_{jj} \geq \tfrac{1}{2}$ *for all $j$. Hence* $\mathbf{P}_t^\top$ *is aperiodic and row-substochastic; on any closed communicating class with no leak (i.e., equality in the column sums), it is row-stochastic.*

Its proof is provided in Appendix B. Having identified $\mathbf{P}_t$ as a lazy *substochastic* diffusion, we unroll the recurrence to obtain an explicit representation of propagated energy. We denote that, for all $k \geq 1$, $\mathbf{E}^{(k)} = \mathbf{P}_t^k \mathbf{E}^{(0)}$. Intuitively, the propagated energy at a node after $k$ steps equals a survival-weighted average of initial energies over $k$-step walks emanating from that node. Assume that there exist $a_{\text{ID}} \leq b_{\text{ID}} < a_{\text{OOD}} \leq b_{\text{OOD}}$ with $\delta := a_{\text{OOD}} - b_{\text{ID}} > 0$ such that $e_i^{(0)} \in [a_{\text{ID}}, b_{\text{ID}}]$ for ID nodes and $e_j^{(0)} \in [a_{\text{OOD}}, b_{\text{OOD}}]$ for OOD nodes (consistent with Eq. 1 used as an OOD score. We now quantify how OOD visitation controls this value as:

**Theorem 5.2** (Energy–OOD linkage). *Define the unnormalized OOD visitation* $\phi_{\text{OOD}}^{(k)}(t) :=$ $\sum_{j\in\mathcal{O}}(\mathbf{P}_t^k)_{tj}$ *and the retained mass* $s_t^{(k)} := \sum_i(\mathbf{P}_t^k)_{ti}$. *For all* $k \geq 1$,

$$a_{\text{ID}}\, s_t^{(k)} + \delta\, \phi_{\text{OOD}}^{(k)}(t) \ \leq\ e_t^{(k)} \ \leq\ b_{\text{ID}}\, s_t^{(k)} + \left(b_{\text{OOD}} - b_{\text{ID}}\right)\phi_{\text{OOD}}^{(k)}(t).$$

*Equivalently, whenever* $s_t^{(k)} > 0$, *with the* conditional *OOD visitation* $\widehat{\pi}_{\text{OOD}}^{(k)}(t) := \phi_{\text{OOD}}^{(k)}(t)/s_t^{(k)}$,

$$a_{\text{ID}} + \delta\, \widehat{\pi}_{\text{OOD}}^{(k)}(t) \ \leq\ \frac{e_t^{(k)}}{s_t^{(k)}} \ \leq\ b_{\text{ID}} + \left(b_{\text{OOD}} - b_{\text{ID}}\right)\widehat{\pi}_{\text{OOD}}^{(k)}(t).$$

The proof is provided in Appendix B. The lower bound increases with slope $\delta > 0$ in the OOD visitation $\phi_{\text{OOD}}^{(k)}$ (or $\widehat{\pi}_{\text{OOD}}^{(k)}$ in conditional form). Therefore, minimizing $\mathcal{L}_{\text{ene}} = e_t^{(k)}$ necessarily reduces OOD visitation along $k$-step walks from $t$. In practice, gradient descent on $\mathcal{L}_{\text{ene}}$ suppresses edges that route mass into high-energy (OOD) regions and retains edges into low-energy (ID) regions, matching the empirical reduction in OOD-edge precision. Lastly, the time complexity of the WEP is given as:

**Lemma 5.3** (Time Complexity). *With sparse matrix–vector multiplies, computing* $\mathbf{E}^{(k)} = \mathbf{P}_t^k\mathbf{E}^{(0)}$ *costs* $O(k|\mathcal{E}|)$ *per epoch; over* $T$ *training epochs,* WEP *runs in* $O(Tk|\mathcal{E}|)$ *time and* $O(|\mathcal{E}|)$ *memory, i.e., linear in the number of edges.*

*Proof.* Each multiplication by $\mathbf{P}_t$ is a sparse matrix with $\mathbf{A}_t^*$ (plus a scaled identity), both $O(|\mathcal{E}|)$. Repeating $k$ times per epoch yields $O(k|\mathcal{E}|)$; with fixed $T, k$ the total is $O(Tk|\mathcal{E}|)$. $\qquad\square$

# 6 EXPERIMENTAL SET-UP

## 6.1 DATASET CONSTRUCTION

We evaluate the proposed ORExplainer with four synthetic datasets and two real-world datasets. The synthetic datasets (BA-Shapes, BA-Community, Tree-Cycles, Tree-Grids) (Ying et al., 2019) are designed to evaluate GNN XAI tasks. For real-world evaluation, we use Cora and Citeseer (Sen et al., 2008), two widely studied citation networks that serve as standard benchmarks for node classification tasks.

To evaluate the explainability methods under OOD conditions, we construct experimental settings that introduce different types of OOD: structure-level OOD, feature-level OOD, unseen-label. Structural OOD involves adding new nodes and edges as noisy OOD instances. In the synthetic datasets, we introduce 10 to 30 new nodes as OOD nodes to measure the impact of their presence. These nodes are connected to the original graph through randomly generated edges, with each node having approximately twice the average degree of the graph. Featural OOD refers to transforming the features of certain nodes into noise. In the real-world datasets, we randomly select approximately 30% of the nodes to act as OOD nodes. The features of these nodes are replaced with noise that contains roughly twice the information content of the original node features. Unseen-abel OOD refers to the addition of nodes with labels that were not present during the GNN training process. Following the setting proposed in Wu et al. (2023b), we simulate the appearance of new labels as OOD instances. In a real-world dataset, the class with the largest number of nodes is designated as the OOD class. We trained a GCN on a modified version of the dataset where all edges connected to OOD nodes were removed, ensuring that no information from OOD nodes influenced the GCN during training. For evaluating the explanations generated by the explainer, we used the graph with OOD nodes restored, which includes the unseen nodes, edges, and labels.

## 6.2 BASELINES

We compare our method with six instance-level post-hoc explainers: GNNExplainer (Ying et al., 2019), PGExplainer (Luo et al., 2020), MixupExplainer (Zhang et al., 2023), ProxyExplainer (Chen et al., 2024), V-InFoR (Wang et al., 2024), HINT-G (Jung et al., 2025). While ProxyExplainer and V-InFoR were originally proposed for graph classification, we adapt them to node classification by extending their edge embedding inputs to include the representations of the two endpoint nodes and the target node. For a fair comparison, we applied the same GNN architecture across all methods.

Table 1: Performance comparison on synthetic datasets with 10 injected struture-level OOD nodes.

| Method | BA-Shapes | | BA-Community | | Tree-Cycle | | Tree-Grid | |
|---|---|---|---|---|---|---|---|---|
| | $AUC$ (↑) | $OOD$ (↓) | $AUC$ (↑) | $OOD$ (↓) | $AUC$ (↑) | $OOD$ (↓) | $AUC$ (↑) | $OOD$ (↓) |
| GNNExplainer | $0.755 \pm 0.006$ | $0.384 \pm 0.014$ | $0.911 \pm 0.004$ | $0.013 \pm 0.003$ | $0.583 \pm 0.014$ | $0.068 \pm 0.011$ | $0.707 \pm 0.001$ | $0.024 \pm 0.001$ |
| PGExplainer | $0.730 \pm 0.062$ | $0.170 \pm 0.013$ | $0.853 \pm 0.028$ | $0.039 \pm 0.006$ | $0.877 \pm 0.013$ | $0.018 \pm 0.001$ | $0.899 \pm 0.014$ | $\mathbf{0.006 \pm 0.002}$ |
| MixupExplainer | $0.766 \pm 0.055$ | $0.151 \pm 0.031$ | $0.858 \pm 0.024$ | $0.035 \pm 0.008$ | $0.884 \pm 0.005$ | $0.019 \pm 0.006$ | $0.897 \pm 0.013$ | $\mathbf{0.006 \pm 0.002}$ |
| ProxyExplainer | $0.732 \pm 0.057$ | $0.148 \pm 0.029$ | $0.851 \pm 0.031$ | $0.037 \pm 0.008$ | $0.884 \pm 0.006$ | $0.018 \pm 0.001$ | $0.897 \pm 0.014$ | $0.007 \pm 0.002$ |
| V-InFoR | $0.501 \pm 0.009$ | $0.034 \pm 0.004$ | $0.554 \pm 0.044$ | $0.040 \pm 0.014$ | $0.515 \pm 0.027$ | $0.066 \pm 0.009$ | $0.498 \pm 0.017$ | $0.071 \pm 0.004$ |
| HINT-G | $0.841 \pm 0.000$ | $0.034 \pm 0.000$ | $0.788 \pm 0.000$ | $0.080 \pm 0.000$ | $0.911 \pm 0.000$ | $0.060 \pm 0.000$ | $0.620 \pm 0.000$ | $0.097 \pm 0.000$ |
| ORExplainer | $\mathbf{0.995 \pm 0.000}$ | $\mathbf{0.017 \pm 0.003}$ | $\mathbf{0.993 \pm 0.000}$ | $\mathbf{0.000 \pm 0.000}$ | $\mathbf{0.954 \pm 0.001}$ | $\mathbf{0.011 \pm 0.000}$ | $\mathbf{0.962 \pm 0.003}$ | $0.007 \pm 0.000$ |

Table 2: Performance comparison on the Real-world datasets with 10% of feature-level OOD nodes assigned noisy features.

| Method | Cora | | | Citeseer | | |
|---|---|---|---|---|---|---|
| | $Fid_+$ (↑) | $Fid_-$ (↓) | $OOD$ (↓) | $Fid_+$ (↑) | $Fid_-$ (↓) | $OOD$ (↓) |
| GNNExplainer | $0.021 \pm 0.002$ | $0.117 \pm 0.002$ | $0.152 \pm 0.006$ | $-0.006 \pm 0.001$ | $0.031 \pm 0.001$ | $0.197 \pm 0.009$ |
| PGExplainer | $0.021 \pm 0.001$ | $0.114 \pm 0.002$ | $0.150 \pm 0.011$ | $0.003 \pm 0.001$ | $0.029 \pm 0.002$ | $0.165 \pm 0.041$ |
| MixupExplainer | $0.020 \pm 0.001$ | $0.118 \pm 0.002$ | $0.138 \pm 0.002$ | $0.004 \pm 0.000$ | $0.028 \pm 0.001$ | $0.147 \pm 0.058$ |
| ProxyExplainer | $0.018 \pm 0.001$ | $0.117 \pm 0.001$ | $0.201 \pm 0.010$ | $0.005 \pm 0.001$ | $0.026 \pm 0.002$ | $0.121 \pm 0.037$ |
| V-InFoR | $0.012 \pm 0.003$ | $0.116 \pm 0.004$ | $0.236 \pm 0.022$ | $0.005 \pm 0.002$ | $0.025 \pm 0.004$ | $0.125 \pm 0.024$ |
| HINT-G | $0.011 \pm 0.000$ | $0.166 \pm 0.000$ | $0.603 \pm 0.000$ | $0.010 \pm 0.000$ | $0.029 \pm 0.000$ | $0.372 \pm 0.000$ |
| ORExplainer | $\mathbf{0.038 \pm 0.001}$ | $\mathbf{0.102 \pm 0.002}$ | $\mathbf{0.037 \pm 0.001}$ | $\mathbf{0.018 \pm 0.001}$ | $\mathbf{0.016 \pm 0.002}$ | $\mathbf{0.005 \pm 0.004}$ |

## 6.3 EVALUATION METRICS

For **synthetic datasets**, where ground truth subgraph motifs are available, we report the Area Under the ROC Curve (AUC) between the generated edge weights and the ground truth explanatory edges. We additionally measure **OOD Edge Precision** (abbreviated as **OOD**), which calculates the fraction of OOD edges contained in the explanatory subgraph. For **real-world datasets**, where ground truth explanations are unavailable, we adopt **Fidelity** (Amara et al., 2022; Yuan et al., 2022), reported in two complementary forms: $Fid_+$ (sufficiency) and $Fid_-$ (necessity). Alongside fidelity, we also report **OOD** to evaluate robustness against OOD nodes and edges.

## 6.4 IMPLEMENTATION DETAILS

We used a 3-layer GCN with a hidden dimension of 20 per layer on the synthetic datasets, and a 2-layer GCN with a hidden dimension of 16 on the real-world datasets. For evaluation, the continuous edge mask is discretized via top-$k, p$ samplings into an explanatory subgraph. In the synthetic datasets, we select the top-$k$ edges, where $k$ matches the number of edges in the ground-truth motif. In the real-world datasets, we instead take the top-$p$ fraction of edges, with $p = 10\%$. Other details of the experimental settings are provided in Appendix C.

## 7 EXPERIMENTAL RESULTS

### 7.1 RESEARCH QUESTION (RQ) 1: QUANTITATIVE EVALUATION

We evaluate the explanations generated by ORExplainer and baseline methods across three representative OOD scenarios: (i) Strucutre-level OOD, (ii) Feature-level OOD, and (iii) Unseen-label OOD. Each scenario highlights a different robustness challenge, and the corresponding results are summarized in Table 1, Table 2, and Table 3, respectively. The full results for all additional OOD settings and datasets are reported in the Appendix D.

Table 1 reports results on synthetic datasets with 10 injected structure-level OOD nodes. ORExplainer consistently achieves the best performance across most datasets, showing the highest AUC while keeping OOD Edge Precision low. This demonstrates that ORExplainer not only identifies the ground-truth explanatory motifs more accurately but also effectively suppresses spurious OOD edges. For the Tree-Grid dataset, ORExplainer records slightly higher OOD values compared to some baselines, but the absolute magnitude remains very small. In contrast, the improvement in AUC is relatively large, indicating that ORExplainer can still capture the true explanatory structure more reliably while being less affected by structural perturbations introduced by OOD nodes. V-InFoR, in contrast, shows low performance since it is originally designed for graph classification and struggles to scale to larger node classification graphs that require effective VGAE training.

Table 3: Performance comparison on real-world datasets where all unseen-label nodes are restored.

| Method | Cora | | | Citeseer | | |
|---|---|---|---|---|---|---|
| | $Fid_+$ ($\uparrow$) | $Fid_-$ ($\downarrow$) | $OOD$ ($\downarrow$) | $Fid_+$ ($\uparrow$) | $Fid_-$ ($\downarrow$) | $OOD$ ($\downarrow$) |
| GNNExplainer | $0.005 \pm 0.001$ | $0.040 \pm 0.001$ | $0.141 \pm 0.006$ | $-0.003 \pm 0.003$ | $0.038 \pm 0.002$ | $0.026 \pm 0.003$ |
| PGExplainer | $0.010 \pm 0.001$ | $0.031 \pm 0.001$ | $0.078 \pm 0.003$ | $0.009 \pm 0.001$ | $0.018 \pm 0.002$ | $0.007 \pm 0.001$ |
| MixupExplainer | $0.010 \pm 0.001$ | $0.032 \pm 0.001$ | $0.079 \pm 0.003$ | $0.010 \pm 0.001$ | $0.018 \pm 0.001$ | $\mathbf{0.005 \pm 0.002}$ |
| ProxyExplainer | $0.010 \pm 0.001$ | $0.033 \pm 0.002$ | $0.074 \pm 0.004$ | $0.009 \pm 0.001$ | $0.019 \pm 0.000$ | $0.008 \pm 0.003$ |
| V-InFoR | $0.001 \pm 0.001$ | $0.039 \pm 0.002$ | $0.174 \pm 0.007$ | $0.005 \pm 0.002$ | $0.032 \pm 0.006$ | $0.033 \pm 0.005$ |
| HINT-G | $-0.002 \pm 0.000$ | $0.059 \pm 0.000$ | $0.174 \pm 0.000$ | $0.005 \pm 0.000$ | $0.049 \pm 0.000$ | $0.008 \pm 0.000$ |
| ORExplainer | $\mathbf{0.020 \pm 0.001}$ | $\mathbf{0.029 \pm 0.001}$ | $\mathbf{0.062 \pm 0.005}$ | $\mathbf{0.026 \pm 0.001}$ | $\mathbf{0.016 \pm 0.002}$ | $0.007 \pm 0.001$ |

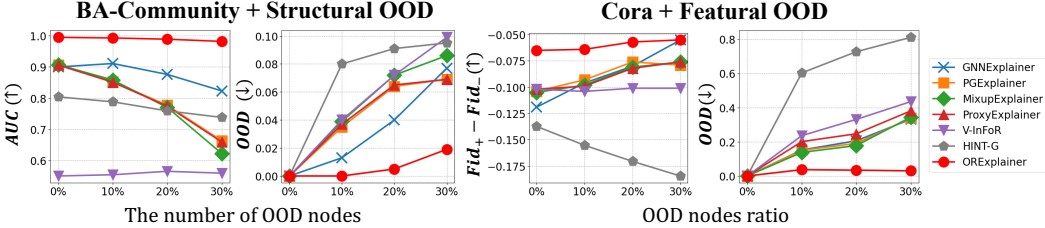

Figure 3: Performance of different explanation methods under varying OOD level

Table 2 presents results on real-world datasets where approximately 10% of nodes have been corrupted with noisy features. ORExplainer consistently outperforms the baselines, achieving the highest $Fid_+$ and lowest $Fid_-$ while also maintaining significantly lower OOD edge precision. In particular, on Citeseer, ORExplainer yields a substantial improvement in $Fid_+$ while keeping the OOD value close to zero, demonstrating that our method can provide stable and ID-focused explanations. In contrast, baselines show higher sensitivity to noisy features, often suffering from increased $Fid_-$ or unstable OOD precision. Since HINT-G solely relies on the trained GNN model without reference to $\mathcal{G}_{\text{explain}}$, unseen OOD nodes or edges yield high influence scores, causing many OOD edges to be included in the extracted explanation subgraph. As a result, edges connected to OOD nodes are frequently selected, inflating OOD precision and undermining the reliability of the resulting explanations.

Similarly, Table 3 reports results in the unseen-label OOD setting, where all previously removed class nodes are restored. ORExplainer again achieves the best overall performance, with consistently higher $Fid_+$ and lower $Fid_-$ across both Cora and Citeseer. On Citeseer, ORExplainer achieves the highest $Fid_+$ among all methods, while keeping the OOD precision at a comparably low level. This indicates that our approach can provide stable and ID-focused explanations even in the presence of unseen-label nodes.

## 7.2 RQ 2: Is ORExplainer robust across various levels of OOD?

This research question investigates whether ORExplainer can maintain robustness under varying levels of OOD across different datasets. Figure 3 presents results on BA-Community (left) and Cora (right), using AUC , the combined fidelity metric ($Fid_+ - Fid_-$), and OOD edge precision (OOD) for evaluation. On both datasets, ORExplainer clearly outperforms all baselines in terms of AUC across different OOD ratios. While PGExplainer, MixupExplainer, and ProxyExplainer exhibit moderate performance at low OOD levels, their scores quickly decline as the ratio increases, showing limited robustness. V-InFoR remains relatively flat, but at a consistently low level, indicating weak explanatory capacity. For the real world dataset Cora, GNNExplainer exhibits high fidelity with the addition of OOD nodes, giving the impression of improved explanatory quality. However, many OOD connected edges are included in the explanations. This compromises the reliability of its explanations, since high fidelity achieved by relying on irrelevant or misleading edges cannot be regarded as trustworthy. By contrast, ORExplainer maintains both high AUC and stable behavior across all OOD levels, demonstrating that it can reliably highlight informative structures without being distracted by OOD nodes.

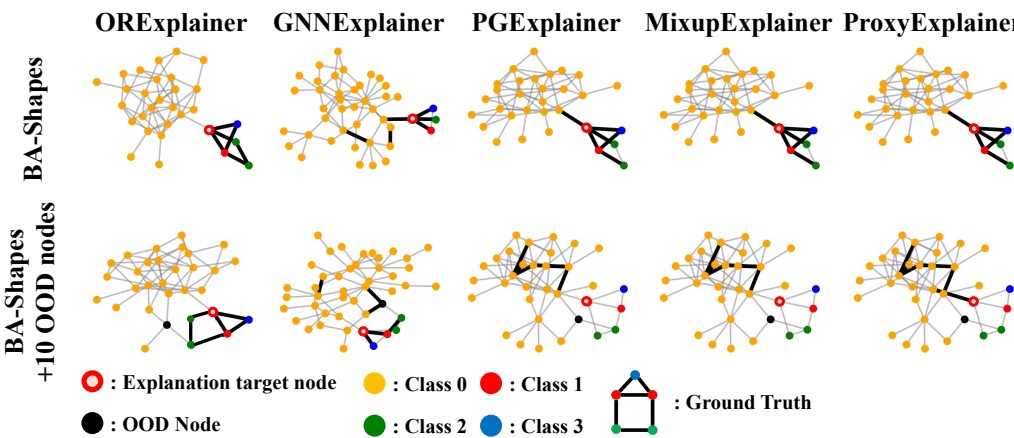

Figure 4: Example explanations generated by different methods on BA-Shapes and BA-Shapes with 10 OOD nodes.

### 7.3 RQ 3: QUALITATIVE ANALYSIS

Figure 4 shows example subgraph explanations for BA-Shapes, comparing cases without and with OOD nodes. The thick black edges indicate those assigned higher weights by each explanation model. When OOD nodes are absent, most methods are able to capture the house motif structure around the target node. However, once OOD nodes and spurious connections are introduced, the baselines frequently highlight irrelevant edges that are disconnected from the underlying motif, reducing the reliability of their explanations. In contrast, ORExplainer consistently assigns high weights to the house motif edges regardless of the presence of OOD nodes, demonstrating its robustness in producing faithful explanations under OOD conditions.

### 7.4 RQ 4: HYPERPARAMETER ANALYSIS

We further investigate the effect of $\gamma$ on BA-Shapes with 30 injected OOD nodes. As shown in Figure 5, when $\gamma$ is small, the performance fluctuates and the variance across runs is relatively large. As $\gamma$ increases, both AUC and OOD precision stabilize, and the standard deviation becomes smaller, indicating that the training process is more stable. This demonstrates that assigning sufficient weight to the robustness term $\mathcal{L}_{\text{ene}}$ allows the explainer to effectively suppress OOD influence and produce consistent explanations.

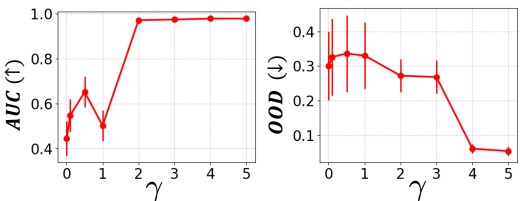

Figure 5: Effect of $\gamma$ on BA-Shapes with 30 OOD nodes. The markers indicate the mean across different random seeds, and the error bars represent the standard deviations.

## 8 CONCLUSION

In this paper, we introduced ORExplainer, a post-hoc, instance-level explanation model designed to provide robust and reliable explanations in graph environments containing out-of-distribution, noisy, and outlier nodes. By incorporating Energy Scores to quantify the GNN's understanding of each node and using the weighted energy score propagation to capture the structural dependencies within the graph, ORExplainer effectively mitigates the impact of OOD nodes while maintaining high explainability for ID nodes. Our extensive experiments demonstrated that existing baseline models are highly sensitive to OOD nodes, resulting in a significant drop in explanation quality and reliability. In contrast, ORExplainer exhibited superior robustness, with smaller performance degradation even as the proportion of OOD nodes increased. These results highlight ORExplainer's ability to provide reliable explanations in real-world graph scenarios where ID and OOD nodes coexist, making it a highly effective tool for GNN interpretability in challenging environments.

## 9 REPRODUCIBILITY STATEMENT

We provide an anonymous GitHub repository containing the implementation and the datasets used in our experiments: https://anonymous.4open.science/r/ORExplainer-C52C. The repository also includes all hyperparameter settings and training scripts. A detailed description of the hyperparameter configurations is additionally provided in Appendix C to further facilitate reproducibility.

## 10 ETHICS STATEMENT

This work does not involve human subjects, personal data, or sensitive information. All experiments are conducted on publicly available benchmark datasets (synthetic datasets and citation networks such as Cora and Citeseer). Our study focuses on developing robust explainability methods for graph neural networks under the presence of out-of-distribution nodes. We do not foresee direct societal harm from the proposed methodology, but we acknowledge that explainability techniques can potentially be misused if applied without consideration of fairness and bias in real-world data. We encourage responsible use of our approach in line with the ICLR Code of Ethics.

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

# A  EXTENDED RELATED WORK

## A.1  GRAPH NEURAL NETWORKS

Graph Neural Networks (GNNs) (Scarselli et al., 2008; Kipf & Welling, 2017) have become fundamental tools for modeling graph-structured data and are widely applied to tasks such as node classification, graph classification, and link prediction. Recent work has explored improving GNN robustness through graph denoising and purification (Li et al., 2024; Lee & Park, 2025), which mitigate the impact of noise, adversarial perturbations, and out-of-distribution (OOD) instances.

## A.2  EXPLAINABILITY IN GRAPH NEURAL NETWORKS

Explainable AI models in the graph domain focus on identifying substructures that significantly impact outputs from trained models. Primarily, GNNExplainer (Ying et al., 2019), a pioneering study in this field, proposes a mask-based method to find important subgraphs that maximize the mutual information with the predictive output. Furthermore, PGExplainer (Luo et al., 2020) advances this concept by parameterizing explainers in a more generalized setting, approximating multiple important subgraphs for various instances using a single explainer. Additionally, SubgraphX (Yuan et al., 2021) employs Monte Carlo Tree Search to identify important subgraphs with the highest Shapley value.

While various state-of-the-art explanation methods contribute to generating high-quality explanations, another line of research questions have emerged regarding their generalization and robustness. MixupExplainer (Zhang et al., 2023) and ProxyExplainer (Chen et al., 2024) address the issue that explanatory subgraphs often suffer from a distribution shift **relative to** the input graphs, due to differences in size or structural properties. Since a pretrained GNN model cannot properly process such distribution-shifted graphs, the training of the explainer itself becomes problematic. To mitigate this problem, MixupExplainer mixes input graphs with label-irrelevant graphs, whereas ProxyExplainer employs a VGAE (Kipf & Welling, 2016) encoder to enforce in-distribution explanations. In a different approach, HINT-G (Jung et al., 2025) leverages influence functions (Bae et al., 2022; Wu et al., 2023a) to trace how training nodes affect the prediction of a target node, providing explanations grounded in influence rather than subgraph generation.

Despite significant advancements in explainability, many existing methods often overlook the impact of OOD nodes and edges that can arise **within** the input graph. V-InFoR (Wang et al., 2024), unlike prior works, focuses on designing a robust explainer for structurally corrupted graphs. It leverages variational inference to learn robust graph representations in order to address structural corruption. However, its robustness mainly targets structure-level OOD and does not extend to other types of corruption, such as feature noise. In addition, since it is originally developed for a graph classification task, its applicability to node-level scenarios such as node injection remains limited.

**Different Setting Compared to Existing Methods:** Most existing explanation methods implicitly assume that the explainer is trained on the same in-distribution graphs as the GNN model. However, real-world graphs are inherently dynamic, continuously evolving through the addition of new nodes and edges. These dynamics naturally introduce out-of-distribution (OOD) components, which existing explainers are not designed to handle. This underscores the necessity of developing explanation methods that are explicitly designed for graphs with newly added OOD nodes or edges.

## A.3  NODE-LEVEL OUT-OF-DISTRIBUTION DETECTION

Node-level OOD detection seeks to distinguish nodes that have a distribution different from the In-Distribution (ID) training data. One popular approach is to train a model for OOD scoring. GPN (Stadler et al., 2021) leverages the Bayesian posterior to train GNNs for uncertainty estimation. OODGAT (Song & Wang, 2022) incorporates entropy regularization alongside GNN training for classification, enabling the distinction between ID and OOD nodes. GraphDE (Li et al., 2022) employs variational inference to identify distributional differences between ID and OOD data. However, these methods have limitations when it comes to scoring OOD nodes based on a pre-trained GNN.

Alternatively, post-hoc OOD detection methods can be applied on a pre-trained classifier. Mahalanobis distance (Lee et al., 2018) utilizes the latent space of a pre-trained classifier to measure the distance between test samples and known in-distribution data, while Kernel Density Estimation (Zhao et al., 2020) models the density of in-distribution samples in the latent space to assess the likelihood of test samples belonging to the same distribution. However, these methods require access to the ID data distribution, which may not always be feasible.

In contrast, logit-based scoring methods such as Maximum Softmax Probability (MSP) (Hendrycks & Gimpel, 2017) and Energy Score (Liu et al., 2020; Wu et al., 2023b) are lightweight and do not require retraining. In particular, prior studies (Wu et al., 2023b; Yang et al., 2024) have shown that energy-based scoring is a simple yet effective baseline for OOD detection across domains, making it especially appealing in settings where the graph to be explained may contain unknown distributions.

# B  EXTENDED THEORETICAL ANALYSIS

**Setup.** Let $\mathbf{E}^{(0)} = [e_1^{(0)}, \ldots, e_N^{(0)}]^\top$ be the initial node energies computed from the fixed GNN $f$ (Eq. 1). WEP (Eq. 3) updates

$$\mathbf{E}^{(k)} = \tfrac{1}{2}\big(\mathbf{E}^{(k-1)} + \mathbf{A}_t^* \mathbf{D}^{-1}\mathbf{E}^{(k-1)}\big) = \mathbf{P}_t\,\mathbf{E}^{(k-1)} = \mathbf{P}_t^k\mathbf{E}^{(0)}, \quad \mathbf{P}_t := \tfrac{1}{2}\big(\mathbf{I} + \mathbf{A}_t^*\mathbf{D}^{-1}\big),$$

where $\mathbf{A}_t^* \in \mathbb{R}_{\geq 0}^{N \times N}$ is the explainer's weighted adjacency for target $t$, and $\mathbf{D} = \mathrm{diag}(d_1, \ldots, d_N)$ is the degree matrix of the *explanation* graph used in Eq. 3. Let $\mathbf{A}_{\mathrm{explain}}$ denote the (binary) adjacency of $G_{\mathrm{explain}}$.

**Assumptions.**

- **A1 (Support & boundedness).** $0 \leq (\mathbf{A}_t^*)_{ij} \leq (\mathbf{A}_{\mathrm{explain}})_{ij}$ element-wise.
- **A2 (Energy gap).** There exist $a_{\mathrm{ID}} \leq b_{\mathrm{ID}} < a_{\mathrm{OOD}} \leq b_{\mathrm{OOD}}$ with $\delta := a_{\mathrm{OOD}} - b_{\mathrm{ID}} > 0$ such that $e_i^{(0)} \in [a_{\mathrm{ID}}, b_{\mathrm{ID}}]$ for ID nodes and $e_j^{(0)} \in [a_{\mathrm{OOD}}, b_{\mathrm{OOD}}]$ for OOD nodes (consistent with Eq. 1 used as an OOD score).
- **A3 (Fixed degree scaling).** $\mathbf{D} = \mathrm{diag}(d_1, \ldots, d_N)$ is formed from $\mathcal{G}_{\mathrm{explain}}$ and does not depend on $\mathbf{A}_t^*$ (as in Eq. 3).

We first establish that the WEP operator $\mathbf{P}_t$ is lazy and *column-substochastic*.

**Lemma 5.1** (Column-substochastic laziness). $\mathbf{P}_t$ satisfies $\sum_i (\mathbf{P}_t)_{ij} \leq 1$ for every $j$, with equality iff $\sum_i (\mathbf{A}_t^*)_{ij} = d_j$, and $(\mathbf{P}_t)_{jj} \geq \tfrac{1}{2}$ for all $j$. Hence $\mathbf{P}_t^\top$ is aperiodic and row-substochastic; on any closed communicating class with no leak (i.e., equality in the column sums), it is row-stochastic.

*Proof.* By definition,

$$\sum_i (\mathbf{P}_t)_{ij} = \tfrac{1}{2}\Big(\sum_i \delta_{ij} + \sum_i (\mathbf{A}_t^*\mathbf{D}^{-1})_{ij}\Big) = \tfrac{1}{2}\Big(1 + \tfrac{1}{d_j}\sum_i (\mathbf{A}_t^*)_{ij}\Big) \leq 1,$$

where the inequality uses A1 and that $d_j = \sum_i (\mathbf{A}_{\mathrm{explain}})_{ij}$. Also $(\mathbf{P}_t)_{jj} = \tfrac{1}{2}\big(1 + (\mathbf{A}_t^*\mathbf{D}^{-1})_{jj}\big) = \tfrac{1}{2}\big(1 + \tfrac{(\mathbf{A}_t^*)_{jj}}{d_j}\big) \geq \tfrac{1}{2}$. The diagonal self-loop probability $\geq \tfrac{1}{2}$ implies aperiodicity for $\mathbf{P}_t^\top$. Equality in the column-sum holds iff $\sum_i (\mathbf{A}_t^*)_{ij} = d_j$. $\square$

Based on the $\mathbf{P}_t^\top$ above, Diffusion representation could be defined as:

**Lemma B.1** (Diffusion representation). *For all $k \geq 1$, $\mathbf{E}^{(k)} = \mathbf{P}_t^k\mathbf{E}^{(0)}$ and, in particular,*

$$e_t^{(k)} = \sum_i (\mathbf{P}_t^k)_{ti}\, e_i^{(0)}.$$

*Proof.* Unroll $\mathbf{E}^{(k)} = \mathbf{P}_t\mathbf{E}^{(k-1)}$ to obtain $\mathbf{E}^{(k)} = \mathbf{P}_t^k\mathbf{E}^{(0)}$. Taking the $t$-th coordinate yields the identity. $\square$

**Theorem 5.2** (Energy–OOD linkage). Define the unnormalized OOD visitation $\phi_{\text{OOD}}^{(k)}(t) := \sum_{j\in\mathcal{O}}(\mathbf{P}_t^k)_{tj}$ and the retained mass $s_t^{(k)} := \sum_i(\mathbf{P}_t^k)_{ti}$. For all $k \geq 1$,

$$a_{\text{ID}}\, s_t^{(k)} + \delta\, \phi_{\text{OOD}}^{(k)}(t) \ \leq\ e_t^{(k)} \ \leq\ b_{\text{ID}}\, s_t^{(k)} + \big(b_{\text{OOD}} - b_{\text{ID}}\big)\, \phi_{\text{OOD}}^{(k)}(t).$$

Equivalently, whenever $s_t^{(k)} > 0$, with the conditional OOD visitation $\widehat{\pi}_{\text{OOD}}^{(k)}(t) := \phi_{\text{OOD}}^{(k)}(t)/s_t^{(k)}$,

$$a_{\text{ID}} + \delta\, \widehat{\pi}_{\text{OOD}}^{(k)}(t) \ \leq\ \frac{e_t^{(k)}}{s_t^{(k)}} \ \leq\ b_{\text{ID}} + \big(b_{\text{OOD}} - b_{\text{ID}}\big)\, \widehat{\pi}_{\text{OOD}}^{(k)}(t).$$

*Proof.* By Lemma B.1,

$$e_t^{(k)} = \sum_{i\in\mathcal{I}}(\mathbf{P}_t^k)_{ti}\, e_i^{(0)} \ + \ \sum_{j\in\mathcal{O}}(\mathbf{P}_t^k)_{tj}\, e_j^{(0)}.$$

**(1) Bound the ID part.** For all $i \in \mathcal{I}$, $a_{\text{ID}} \leq e_i^{(0)} \leq b_{\text{ID}}$, hence

$$a_{\text{ID}}\sum_{i\in\mathcal{I}}(\mathbf{P}_t^k)_{ti} \ \leq\ \sum_{i\in\mathcal{I}}(\mathbf{P}_t^k)_{ti}e_i^{(0)} \ \leq\ b_{\text{ID}}\sum_{i\in\mathcal{I}}(\mathbf{P}_t^k)_{ti}.$$

Since $\sum_{i\in\mathcal{I}}(\mathbf{P}_t^k)_{ti} = s_t^{(k)} - \phi_{\text{OOD}}^{(k)}(t)$, this becomes

$$a_{\text{ID}}\big(s_t^{(k)} - \phi_{\text{OOD}}^{(k)}(t)\big) \ \leq\ \sum_{i\in\mathcal{I}}(\mathbf{P}_t^k)_{ti}e_i^{(0)} \ \leq\ b_{\text{ID}}\big(s_t^{(k)} - \phi_{\text{OOD}}^{(k)}(t)\big).$$

**(2) Bound the OOD part.** For all $j \in \mathcal{O}$, $a_{\text{OOD}} \leq e_j^{(0)} \leq b_{\text{OOD}}$, hence

$$a_{\text{OOD}}\, \phi_{\text{OOD}}^{(k)}(t) \ \leq\ \sum_{j\in\mathcal{O}}(\mathbf{P}_t^k)_{tj}e_j^{(0)} \ \leq\ b_{\text{OOD}}\, \phi_{\text{OOD}}^{(k)}(t).$$

**(3) Add the bounds.** Summing yields

$$a_{\text{ID}}\big(s_t^{(k)} - \phi_{\text{OOD}}^{(k)}\big) + a_{\text{OOD}}\phi_{\text{OOD}}^{(k)} \ \leq\ e_t^{(k)} \ \leq\ b_{\text{ID}}\big(s_t^{(k)} - \phi_{\text{OOD}}^{(k)}\big) + b_{\text{OOD}}\phi_{\text{OOD}}^{(k)},$$

where we abbreviate $\phi_{\text{OOD}}^{(k)} = \phi_{\text{OOD}}^{(k)}(t)$. Rearranging and substituting $a_{\text{OOD}} = b_{\text{ID}} + \delta$ gives the first display; dividing by $s_t^{(k)}$ (when $s_t^{(k)} > 0$) yields the conditional statement. $\square$

Theorem 5.2 above yields an explicit upper bound on the OOD visitation in terms of the propagated energy. Rearranging the lower bound gives

$$\phi_{\text{OOD}}^{(k)}(t) \ \leq\ \frac{e_t^{(k)} - a_{\text{ID}}\, s_t^{(k)}}{\delta}. \tag{6}$$

Consequently, if during training we enforce $e_t^{(k)} \leq \tau$ for some threshold $\tau > 0$, then

$$\phi_{\text{OOD}}^{(k)}(t) \ \leq\ \frac{\tau - a_{\text{ID}}\, s_t^{(k)}}{\delta}. \tag{7}$$

For fixed retained mass $s_t^{(k)}$ and energy gap $\delta$, the WEP regularizer $\mathcal{L}_{\text{ene}}$ directly upper-bounds the total probability mass of $k$-step walks from $t$ that ever visit OOD nodes. In this sense, $\mathcal{L}_{\text{ene}}$ is a quantitative surrogate for constraining path-based OOD exposure, which is empirically reflected in the reduced OOD edge precision reported in Section 7.

So far we have established how $\mathcal{L}_{\text{ene}}$ controls robustness to OOD nodes. We next clarify how the cross-entropy term in Eq. 4 formalizes faithfulness of the explanation. Let $p_t := f(\mathcal{G}_{\text{explain}}, t)$ and $q_t := f(\mathcal{G}_t^*, t)$ denote the predictive class distributions (after softmax) of the pre-trained GNN on the full graph $\mathcal{G}_{\text{explain}}$ and on the explanatory subgraph $\mathcal{G}_t^*$, respectively. The cross-entropy loss can be written as

$$\mathcal{L}_{\text{CE}} = \text{CE}(p_t, q_t) = H(p_t) + \text{KL}\big(p_t \,\|\, q_t\big), \tag{8}$$

where $H(\cdot)$ is the Shannon entropy and $\text{KL}(\cdot\|\cdot)$ is the Kullback–Leibler divergence. Since $p_t$ is fixed by the pre-trained GNN and the input graph, $H(p_t)$ is constant with respect to the explainer

parameters, so minimizing $\mathcal{L}_{\text{CE}}$ is equivalent to minimizing $\text{KL}(p_t \| q_t)$. Thus, the cross-entropy term encourages the explanatory subgraph to preserve the original predictive distribution on the target node up to small KL divergence, providing an information-theoretic notion of faithfulness that complements the robustness control offered by $\mathcal{L}_{\text{ene}}$. Together, the composite objective in Eq. 4 couples a surrogate control of OOD exposure with a distributional matching term for faithfulness.

In summary, under A1–A3, WEP forms a lazy *substochastic* diffusion whose propagated energy equals a $k$-step survival-weighted average of initial energies (Lemmas 5.1–B.1). Moreover, the target energy is tightly bounded by OOD visitation, so minimizing $\mathcal{L}_{\text{ene}}$ suppresses OOD exposure (Theorem 5.2).

## C  EXPERIMENTAL SETTINGS

### C.1  EVALUATION METRICS

We evaluated it using Fidelity Amara et al. (2022); Yuan et al. (2022), a commonly used metric in the XAI field. Fidelity ($Fid$) is a metric that evaluates the quality of an explanation by measuring how well the explanatory subgraph supports the model's prediction. It consists of two complementary components: $Fid_+$ and $Fid_-$. A higher $Fid_+$ indicates that the explanatory subgraph contains sufficient information to retain the model's prediction for the class $\hat{y}_t$. In contrast, a lower $Fid_-$ suggests that the explanatory subgraph contains necessary information for the model's prediction, meaning that removing the explanatory subgraph significantly impacts the prediction. $Fidelities$ are defined as follows:

$$Fid_+ = f(\mathcal{G}_{\text{explain}}, t)_{[\hat{y}_t]} - f((\mathcal{G}_{\text{explain}} - \mathcal{G}_t^*), t)_{[\hat{y}_t]}, \tag{9}$$

$$Fid_- = f(\mathcal{G}_{\text{explain}}, t)_{[\hat{y}_t]} - f(\mathcal{G}_t^*, t)_{[\hat{y}_t]}, \tag{10}$$

where $\hat{y}_t = \arg\max_c f(\mathcal{G}_{\text{explain}}, t)_{[c]}$. $f(\mathcal{G}_{\text{explain}}, t)_{[\hat{y}_t]}$ denotes the predicted probability assigned by the pre-trained GNN $f$ to class $\hat{y}_t$ on the target node $v_t$. The explanatory subgraph $\mathcal{G}_t^*$ is generated by the explainer for $v_t$ within $\mathcal{G}_{\text{explain}}$. Since the adjacency matrix of $\mathcal{G}_t^*$ is continuous, it is discretized via top-$k$ or top-$p$ sampling as described above. $\mathcal{G}_{\text{explain}} - \mathcal{G}_t^*$ denotes the graph obtained by removing all edges of the explanatory subgraph $\mathcal{G}_t^*$ from the input graph $\mathcal{G}_{\text{explain}}$.

### C.2  GNN TRAINING

Table 4: GNN model and training parameters

| Dataset | Synthetic | Cora, Citeseer |
|---|---|---|
| Layer | 3 | 2 |
| Hidden dimension | 20 | 16 |
| Epochs | 1000 | 200 |
| Learning rate | 0.001 | 0.01 |
| Weight decay | $5 \times 10^{-3}$ | $5 \times 10^{-4}$ |
| Dropout | 0 | 0.05 |
| Embedding concat | Yes | No |

Table 4 shows the hyperparameters when we train the GNN model. We utilize the Adam optimizer. The term Embedding concat refers to constructing node representations by concatenating the intermediate embeddings from all GNN layers together. For synthetic datasets, we adopt an 8:1:1 split ratio for training, validation, and test sets, respectively. For real-world datasets, we follow the standard semi-supervised setting. The GNN model is trained on graphs where all OOD nodes have been removed. For a given dataset, the same GNN model is explained regardless of the OOD level.

In Table 5, the OOD level corresponds to the number of structure-level OOD nodes for the synthetic datasets, While for Cora and Citeseer, it refers to the ratio of feature-level OOD nodes. For all experiments, we ensure that explanations are generated only for nodes whose predictions by the GNN remained correct after OOD nodes were added.

| OOD Level | BA-Shapes | BA-Comm. | Tree-Cycle | Tree-Grid | Cora | Citeseer |
|:---:|:---:|:---:|:---:|:---:|:---:|:---:|
| 0 | 0.986 | 0.786 | 0.977 | 0.984 | 0.766 | 0.680 |
| 10 | 0.957 | 0.793 | 0.943 | 0.976 | 0.763 | 0.677 |
| 20 | 0.943 | 0.764 | 0.966 | 0.976 | 0.766 | 0.673 |
| 30 | 0.886 | 0.771 | 0.955 | 0.952 | 0.763 | 0.669 |

| Unseen-label OOD | Cora | Citeseer |
|:---:|:---:|:---:|
| without | 0.746 | 0.784 |
| with | 0.741 | 0.780 |

Table 5: GNN test accuracy under different OOD settings.

## C.3   BASELINE TRAINING

For synthetic datasets, we applied the same hyperparameter settings as reported in the official implementations of each baseline explainer. For real-world datasets, we tune hyperparameters within the following search space.

Table 6: Hyperparameter search ranges for baselines

| Method | Learning rate | Epochs | *Size* | *Entropy* | Others |
|---|---|---|---|---|---|
| GNNExplainer | $[0.01, 0.1]$ | $[10, 100]$ | $[0.001, 0.01]$ | $[0.1, 1.0]$ | |
| PGExplainer | $[0.001, 0.01]$ | $[10, 100]$ | $[0.001, 1.0]$ | $[10^{-4}, 1.0]$ | |
| MixupExplainer | $[0.001, 0.01]$ | $[10, 100]$ | $[0.001, 1.0]$ | $[10^{-4}, 1.0]$ | |
| ProxyExplainer | $[0.001, 0.01]$ | $[10, 100]$ | $[0.001, 1.0]$ | $[10^{-4}, 1.0]$ | |
| V-InFoR | $[0.001, 0.01]$ | $[10, 100]$ | | | $\beta \in [0.1, 1.0], \pi \in [0.1, 1.0],$ $\tau \in [0.1, 0.5]$ |

Table 6 shows the hyperparameter search space of the baselines. Here, *Size* and *Entropy* correspond to the $\ell_1$ size regularizer and the entropy term used to control the explanation mask, respectively. V-InFoR involves different hyperparameters, which are listed separately. HINT-G is a training-free model, and thus, no additional hyperparameter search is conducted.

## C.4   OREXPLAINER TRAINING

Table 7 summarizes the hyperparameter settings used for the experiments of ORExplainer. For the synthetic datasets, the learning rate, number of epochs, $\alpha$, and $\beta$ were set according to the PGExplainer implementation[1], since ORExplainer employs an MLP architecture similar to that of PGExplainer, which ensures a fair comparison with other mask-based methods.

Table 7: Hyperparameter search ranges (in brackets) and the selected values (in bold) for ORExplainer across different datasets and OOD types.

| OOD type | Dataset | Learning rate | Epochs | $\alpha$ | $\beta$ | $\gamma$ |
|---|---|---|---|---|---|---|
| Structural | BA-Shapes | **0.003** | **10** | **0.05** | **1.0** | $[0.1, 5.0]$, **5.0** |
| | BA-Community | **0.003** | **20** | **0.05** | **1.0** | $[0.1, 5.0]$, **5.0** |
| | Tree-Cycle | **0.003** | **20** | **0.1** | **1.0** | $[0.1, 5.0]$, **5.0** |
| | Tree-Grid | **0.003** | **30** | **1.0** | **1.0** | $[0.1, 10.0]$, **10.0** |
| Featural | Cora | $[0.001, 0.1]$, **0.005** | $[10, 100]$, **20** | $[0.1, 1.0]$, **1.0** | $[10^{-4}, 0.1]$, $\mathbf{5 \times 10^{-4}}$ | $[10^{-3}, 0.5]$, **0.1** |
| | Citeseer | $[0.001, 0.1]$, **0.005** | $[10, 100]$, **20** | $[0.1, 1.0]$, **1.0** | $[10^{-4}, 0.1]$, $\mathbf{5 \times 10^{-4}}$ | $[10^{-3}, 0.5]$, **0.1** |
| Unseen | Cora | $[0.001, 0.1]$, **0.005** | $[10, 100]$, **20** | $[0.1, 1.0]$, **1.0** | $[10^{-4}, 0.1]$, $\mathbf{5 \times 10^{-4}}$ | $[10^{-3}, 0.5]$, **0.1** |
| | Citeseer | $[0.001, 0.1]$, **0.005** | $[10, 100]$, **20** | $[0.1, 1.0]$, **1.0** | $[10^{-4}, 0.1]$, $\mathbf{5 \times 10^{-4}}$ | $[10^{-3}, 0.5]$, **0.05** |

---

[1] https://github.com/LarsHoldijk/RE-ParameterizedExplainerForGraphNeuralNetworks

## D  EXTENDED EXPERIMENTAL RESULTS

Table 8, Table 9, and Table 10 present additional results under varying levels of structure-level, feature-level, and unseen-label OOD settings, respectively. Across all scenarios, ORExplainer consistently outperforms baselines in terms of both AUC and fidelity, while also selecting fewer OOD edges. These results demonstrate that ORExplainer produces more reliable explanations by focusing on in-distribution structure even under different OOD levels.

Table 8: Performance comparison on synthetic datasets (BA-Shapes, BA-Community, Tree-Cycle, Tree-Grid) with different numbers of OOD nodes (0, 10, 20, 30). Reported are the mean $\pm$ standard deviation for $AUC$ and $OOD$ ratio.

| # OOD | Method | BA-Shapes $AUC$ ($\uparrow$) | BA-Shapes $OOD$ ($\downarrow$) | BA-Community $AUC$ ($\uparrow$) | BA-Community $OOD$ ($\downarrow$) | Tree-Cycle $AUC$ ($\uparrow$) | Tree-Cycle $OOD$ ($\downarrow$) | Tree-Grid $AUC$ ($\uparrow$) | Tree-Grid $OOD$ ($\downarrow$) |
|---|---|---|---|---|---|---|---|---|---|
| 0 | GNNExplainer | $0.785 \pm 0.010$ | $0.000 \pm 0.000$ | $0.900 \pm 0.004$ | $0.000 \pm 0.000$ | $0.559 \pm 0.010$ | $0.000 \pm 0.000$ | $0.661 \pm 0.002$ | $0.000 \pm 0.000$ |
| | PGExplainer | $0.956 \pm 0.016$ | $0.000 \pm 0.000$ | $0.906 \pm 0.020$ | $0.000 \pm 0.000$ | $0.896 \pm 0.009$ | $0.000 \pm 0.000$ | $0.900 \pm 0.030$ | $0.000 \pm 0.000$ |
| | MixupExplainer | $0.913 \pm 0.093$ | $0.000 \pm 0.000$ | $0.907 \pm 0.016$ | $0.000 \pm 0.000$ | $0.909 \pm 0.004$ | $0.000 \pm 0.000$ | $0.900 \pm 0.030$ | $0.000 \pm 0.000$ |
| | ProxyExplainer | $0.961 \pm 0.013$ | $0.000 \pm 0.000$ | $0.906 \pm 0.020$ | $0.000 \pm 0.000$ | $0.906 \pm 0.004$ | $0.000 \pm 0.000$ | $0.899 \pm 0.030$ | $0.000 \pm 0.000$ |
| | V-InFoR | $0.502 \pm 0.017$ | $0.000 \pm 0.000$ | $0.550 \pm 0.035$ | $0.000 \pm 0.000$ | $0.514 \pm 0.020$ | $0.000 \pm 0.000$ | $0.492 \pm 0.010$ | $0.000 \pm 0.000$ |
| | HINT-G | $0.910 \pm 0.000$ | $0.000 \pm 0.000$ | $0.804 \pm 0.000$ | $0.000 \pm 0.000$ | $\mathbf{0.976 \pm 0.000}$ | $0.000 \pm 0.000$ | $0.819 \pm 0.000$ | $0.000 \pm 0.000$ |
| | ORExplainer | $\mathbf{0.999 \pm 0.000}$ | $0.000 \pm 0.000$ | $\mathbf{0.995 \pm 0.000}$ | $0.000 \pm 0.000$ | $0.950 \pm 0.027$ | $0.000 \pm 0.000$ | $\mathbf{0.990 \pm 0.000}$ | $\mathbf{0.000 \pm 0.000}$ |
| 10 | GNNExplainer | $0.755 \pm 0.012$ | $0.384 \pm 0.014$ | $0.911 \pm 0.004$ | $0.014 \pm 0.003$ | $0.583 \pm 0.014$ | $0.068 \pm 0.011$ | $0.707 \pm 0.001$ | $0.024 \pm 0.001$ |
| | PGExplainer | $0.730 \pm 0.062$ | $0.151 \pm 0.031$ | $0.853 \pm 0.028$ | $0.035 \pm 0.008$ | $0.877 \pm 0.013$ | $0.019 \pm 0.006$ | $0.899 \pm 0.014$ | $\mathbf{0.006 \pm 0.002}$ |
| | MixupExplainer | $0.766 \pm 0.055$ | $0.170 \pm 0.013$ | $0.858 \pm 0.024$ | $0.039 \pm 0.006$ | $0.884 \pm 0.005$ | $0.018 \pm 0.001$ | $0.897 \pm 0.013$ | $0.007 \pm 0.002$ |
| | ProxyExplainer | $0.732 \pm 0.057$ | $0.148 \pm 0.029$ | $0.851 \pm 0.031$ | $0.037 \pm 0.008$ | $0.884 \pm 0.006$ | $0.018 \pm 0.001$ | $0.897 \pm 0.014$ | $\mathbf{0.006 \pm 0.002}$ |
| | V-InFoR | $0.501 \pm 0.009$ | $0.034 \pm 0.004$ | $0.554 \pm 0.044$ | $0.040 \pm 0.014$ | $0.515 \pm 0.027$ | $0.066 \pm 0.009$ | $0.498 \pm 0.017$ | $0.071 \pm 0.004$ |
| | HINT-G | $0.841 \pm 0.000$ | $0.134 \pm 0.000$ | $0.788 \pm 0.000$ | $0.080 \pm 0.000$ | $0.911 \pm 0.000$ | $0.060 \pm 0.000$ | $0.620 \pm 0.000$ | $0.097 \pm 0.000$ |
| | ORExplainer | $\mathbf{0.995 \pm 0.000}$ | $0.017 \pm 0.000$ | $\mathbf{0.993 \pm 0.000}$ | $0.011 \pm 0.000$ | $\mathbf{0.954 \pm 0.001}$ | $0.011 \pm 0.000$ | $\mathbf{0.962 \pm 0.000}$ | $0.007 \pm 0.000$ |
| 20 | GNNExplainer | $0.680 \pm 0.012$ | $0.607 \pm 0.008$ | $0.876 \pm 0.006$ | $0.040 \pm 0.006$ | $0.602 \pm 0.007$ | $0.098 \pm 0.007$ | $0.728 \pm 0.001$ | $0.039 \pm 0.001$ |
| | PGExplainer | $0.490 \pm 0.085$ | $0.197 \pm 0.074$ | $0.777 \pm 0.030$ | $0.064 \pm 0.017$ | $0.870 \pm 0.011$ | $0.041 \pm 0.010$ | $0.888 \pm 0.010$ | $0.015 \pm 0.003$ |
| | MixupExplainer | $0.509 \pm 0.079$ | $0.215 \pm 0.077$ | $0.770 \pm 0.029$ | $0.072 \pm 0.013$ | $0.877 \pm 0.006$ | $0.035 \pm 0.003$ | $0.887 \pm 0.010$ | $0.016 \pm 0.002$ |
| | ProxyExplainer | $0.493 \pm 0.085$ | $0.197 \pm 0.073$ | $0.774 \pm 0.033$ | $0.065 \pm 0.018$ | $0.877 \pm 0.005$ | $0.034 \pm 0.003$ | $0.887 \pm 0.010$ | $0.015 \pm 0.003$ |
| | V-InFoR | $0.497 \pm 0.014$ | $0.068 \pm 0.014$ | $0.565 \pm 0.034$ | $0.072 \pm 0.030$ | $0.511 \pm 0.021$ | $0.124 \pm 0.008$ | $0.502 \pm 0.016$ | $0.123 \pm 0.006$ |
| | HINT-G | $0.791 \pm 0.000$ | $0.131 \pm 0.000$ | $0.759 \pm 0.000$ | $0.091 \pm 0.000$ | $0.885 \pm 0.000$ | $0.103 \pm 0.000$ | $0.617 \pm 0.000$ | $0.135 \pm 0.000$ |
| | ORExplainer | $\mathbf{0.989 \pm 0.000}$ | $0.018 \pm 0.000$ | $\mathbf{0.989 \pm 0.000}$ | $0.005 \pm 0.001$ | $\mathbf{0.947 \pm 0.001}$ | $0.015 \pm 0.000$ | $\mathbf{0.934 \pm 0.002}$ | $0.011 \pm 0.001$ |
| 30 | GNNExplainer | $0.646 \pm 0.011$ | $0.682 \pm 0.010$ | $0.823 \pm 0.005$ | $0.077 \pm 0.006$ | $0.620 \pm 0.013$ | $0.087 \pm 0.010$ | $0.728 \pm 0.001$ | $0.055 \pm 0.001$ |
| | PGExplainer | $0.444 \pm 0.077$ | $0.319 \pm 0.102$ | $0.622 \pm 0.035$ | $0.086 \pm 0.011$ | $0.852 \pm 0.006$ | $0.041 \pm 0.002$ | $0.887 \pm 0.009$ | $0.021 \pm 0.005$ |
| | MixupExplainer | $0.457 \pm 0.079$ | $0.319 \pm 0.102$ | $0.622 \pm 0.035$ | $0.086 \pm 0.011$ | $0.852 \pm 0.006$ | $0.041 \pm 0.002$ | $0.886 \pm 0.009$ | $0.021 \pm 0.005$ |
| | ProxyExplainer | $0.447 \pm 0.080$ | $0.304 \pm 0.104$ | $0.659 \pm 0.047$ | $0.069 \pm 0.008$ | $0.852 \pm 0.007$ | $0.042 \pm 0.003$ | $0.886 \pm 0.009$ | $0.019 \pm 0.004$ |
| | V-InFoR | $0.519 \pm 0.023$ | $0.080 \pm 0.021$ | $0.559 \pm 0.031$ | $0.099 \pm 0.033$ | $0.496 \pm 0.018$ | $0.129 \pm 0.023$ | $0.502 \pm 0.012$ | $0.148 \pm 0.008$ |
| | HINT-G | $0.712 \pm 0.000$ | $0.224 \pm 0.000$ | $0.738 \pm 0.000$ | $0.095 \pm 0.000$ | $0.882 \pm 0.000$ | $0.104 \pm 0.000$ | $0.614 \pm 0.000$ | $0.172 \pm 0.000$ |
| | ORExplainer | $\mathbf{0.978 \pm 0.000}$ | $0.054 \pm 0.013$ | $\mathbf{0.982 \pm 0.003}$ | $0.019 \pm 0.004$ | $\mathbf{0.934 \pm 0.001}$ | $0.034 \pm 0.000$ | $\mathbf{0.906 \pm 0.004}$ | $0.015 \pm 0.002$ |

Table 9: Performance comparison across different OOD ratios (0%, 10%, 20%, 30%) on Cora and Citeseer. Reported are mean $\pm$ standard deviation for Fidelity ($Fid_+$, $Fid_-$) and OOD ratio.

| OOD Ratio | Method | Cora $Fid_+$ ($\uparrow$) | Cora $Fid_-$ ($\downarrow$) | Cora $OOD$ ($\downarrow$) | Citeseer $Fid_+$ ($\uparrow$) | Citeseer $Fid_-$ ($\downarrow$) | Citeseer $OOD$ ($\downarrow$) |
|---|---|---|---|---|---|---|---|
| 0% | GNNExplainer | $0.010 \pm 0.003$ | $0.129 \pm 0.006$ | $0.000 \pm 0.000$ | $-0.005 \pm 0.001$ | $0.036 \pm 0.002$ | $0.000 \pm 0.000$ |
| | PGExplainer | $0.018 \pm 0.001$ | $0.122 \pm 0.002$ | $0.000 \pm 0.000$ | $0.002 \pm 0.001$ | $0.035 \pm 0.001$ | $0.000 \pm 0.000$ |
| | MixupExplainer | $0.018 \pm 0.001$ | $0.123 \pm 0.001$ | $0.000 \pm 0.000$ | $0.002 \pm 0.001$ | $0.034 \pm 0.001$ | $0.000 \pm 0.000$ |
| | ProxyExplainer | $0.019 \pm 0.002$ | $0.121 \pm 0.002$ | $0.000 \pm 0.000$ | $0.002 \pm 0.001$ | $0.036 \pm 0.001$ | $0.000 \pm 0.000$ |
| | V-InFoR | $0.012 \pm 0.003$ | $0.114 \pm 0.009$ | $0.000 \pm 0.000$ | $0.005 \pm 0.003$ | $0.029 \pm 0.005$ | $0.000 \pm 0.000$ |
| | HINT-G | $0.007 \pm 0.000$ | $0.144 \pm 0.000$ | $0.000 \pm 0.000$ | $0.002 \pm 0.000$ | $0.028 \pm 0.000$ | $0.000 \pm 0.000$ |
| | ORExplainer | $\mathbf{0.038 \pm 0.001}$ | $\mathbf{0.103 \pm 0.003}$ | $0.000 \pm 0.000$ | $\mathbf{0.016 \pm 0.002}$ | $\mathbf{0.024 \pm 0.002}$ | $0.000 \pm 0.000$ |
| 10% | GNNExplainer | $0.021 \pm 0.002$ | $0.117 \pm 0.002$ | $0.152 \pm 0.006$ | $-0.006 \pm 0.001$ | $0.031 \pm 0.001$ | $0.197 \pm 0.005$ |
| | PGExplainer | $0.021 \pm 0.001$ | $0.114 \pm 0.002$ | $0.150 \pm 0.011$ | $0.003 \pm 0.001$ | $0.029 \pm 0.002$ | $0.165 \pm 0.041$ |
| | MixupExplainer | $0.020 \pm 0.001$ | $0.118 \pm 0.002$ | $0.138 \pm 0.007$ | $0.004 \pm 0.000$ | $0.028 \pm 0.001$ | $0.147 \pm 0.056$ |
| | ProxyExplainer | $0.018 \pm 0.001$ | $0.117 \pm 0.001$ | $0.201 \pm 0.010$ | $0.005 \pm 0.001$ | $0.026 \pm 0.002$ | $0.121 \pm 0.033$ |
| | V-InFoR | $0.012 \pm 0.003$ | $0.116 \pm 0.004$ | $0.236 \pm 0.022$ | $0.005 \pm 0.002$ | $0.025 \pm 0.004$ | $0.125 \pm 0.024$ |
| | HINT-G | $0.011 \pm 0.000$ | $0.166 \pm 0.000$ | $0.603 \pm 0.000$ | $0.010 \pm 0.000$ | $0.029 \pm 0.000$ | $0.372 \pm 0.000$ |
| | ORExplainer | $\mathbf{0.038 \pm 0.001}$ | $\mathbf{0.102 \pm 0.002}$ | $\mathbf{0.037 \pm 0.001}$ | $\mathbf{0.018 \pm 0.001}$ | $\mathbf{0.016 \pm 0.002}$ | $\mathbf{0.005 \pm 0.000}$ |
| 20% | GNNExplainer | $0.024 \pm 0.004$ | $0.104 \pm 0.004$ | $0.206 \pm 0.005$ | $-0.006 \pm 0.001$ | $0.026 \pm 0.002$ | $0.287 \pm 0.011$ |
| | PGExplainer | $0.027 \pm 0.000$ | $0.103 \pm 0.001$ | $0.191 \pm 0.001$ | $0.002 \pm 0.001$ | $0.025 \pm 0.001$ | $0.231 \pm 0.033$ |
| | MixupExplainer | $0.025 \pm 0.001$ | $0.106 \pm 0.001$ | $0.178 \pm 0.003$ | $0.001 \pm 0.001$ | $0.026 \pm 0.001$ | $0.272 \pm 0.003$ |
| | ProxyExplainer | $0.022 \pm 0.002$ | $0.104 \pm 0.003$ | $0.247 \pm 0.013$ | $0.006 \pm 0.002$ | $0.022 \pm 0.003$ | $0.161 \pm 0.022$ |
| | V-InFoR | $0.010 \pm 0.004$ | $0.111 \pm 0.006$ | $0.331 \pm 0.010$ | $0.005 \pm 0.002$ | $0.021 \pm 0.003$ | $0.230 \pm 0.017$ |
| | HINT-G | $0.019 \pm 0.000$ | $0.190 \pm 0.000$ | $0.727 \pm 0.000$ | $0.008 \pm 0.000$ | $0.025 \pm 0.000$ | $0.540 \pm 0.000$ |
| | ORExplainer | $\mathbf{0.040 \pm 0.002}$ | $\mathbf{0.097 \pm 0.001}$ | $\mathbf{0.034 \pm 0.003}$ | $\mathbf{0.019 \pm 0.001}$ | $\mathbf{0.011 \pm 0.001}$ | $\mathbf{0.007 \pm 0.001}$ |
| 30% | GNNExplainer | $0.034 \pm 0.003$ | $\mathbf{0.089 \pm 0.003}$ | $0.332 \pm 0.004$ | $-0.006 \pm 0.001$ | $0.024 \pm 0.001$ | $0.422 \pm 0.010$ |
| | PGExplainer | $0.022 \pm 0.001$ | $0.101 \pm 0.001$ | $0.339 \pm 0.011$ | $0.003 \pm 0.001$ | $0.020 \pm 0.001$ | $0.347 \pm 0.038$ |
| | MixupExplainer | $0.025 \pm 0.001$ | $0.101 \pm 0.002$ | $0.344 \pm 0.003$ | $0.003 \pm 0.001$ | $0.020 \pm 0.000$ | $0.348 \pm 0.029$ |
| | ProxyExplainer | $0.020 \pm 0.001$ | $0.096 \pm 0.002$ | $0.381 \pm 0.005$ | $0.005 \pm 0.002$ | $0.018 \pm 0.002$ | $0.229 \pm 0.007$ |
| | V-InFoR | $0.010 \pm 0.003$ | $0.111 \pm 0.011$ | $0.435 \pm 0.007$ | $0.004 \pm 0.000$ | $0.018 \pm 0.004$ | $0.315 \pm 0.027$ |
| | HINT-G | $0.010 \pm 0.000$ | $0.195 \pm 0.000$ | $0.811 \pm 0.000$ | $0.012 \pm 0.000$ | $0.046 \pm 0.000$ | $0.576 \pm 0.000$ |
| | ORExplainer | $\mathbf{0.037 \pm 0.001}$ | $0.092 \pm 0.002$ | $\mathbf{0.030 \pm 0.001}$ | $\mathbf{0.019 \pm 0.001}$ | $\mathbf{0.005 \pm 0.001}$ | $\mathbf{0.007 \pm 0.000}$ |

### D.1  RUNTIME ANALYSIS

The results in Table 11 are obtained on BA-Community with 30 structure-level OOD nodes and on Citeseer with a 30% feature-level OOD ratio. The runtime is reported in seconds per node. Among

Table 10: Performance comparison on Cora and Citeseer with and without unseen label nodes. Reported are mean $\pm$ standard deviation for Fidelity ($Fid_+$, $Fid_-$) and OOD ratio across different explainers.

| | Method | Cora | | | Citeseer | | |
|---|---|---|---|---|---|---|---|
| | | $Fid_+$ ($\uparrow$) | $Fid_-$ ($\downarrow$) | $OOD$ ($\downarrow$) | $Fid_+$ ($\uparrow$) | $Fid_-$ ($\downarrow$) | $OOD$ ($\downarrow$) |
| witout unseen label nodes | GNNExplainer | $0.001 \pm 0.002$ | $0.048 \pm 0.001$ | $0.000 \pm 0.000$ | $-0.003 \pm 0.002$ | $0.047 \pm 0.002$ | $0.000 \pm 0.000$ |
| | PGExplainer | $0.007 \pm 0.001$ | $0.034 \pm 0.001$ | $0.000 \pm 0.000$ | $0.018 \pm 0.001$ | $0.031 \pm 0.003$ | $0.000 \pm 0.000$ |
| | MixupExplainer | $0.006 \pm 0.001$ | $0.035 \pm 0.000$ | $0.000 \pm 0.000$ | $0.010 \pm 0.001$ | $0.026 \pm 0.002$ | $0.000 \pm 0.000$ |
| | ProxyExplainer | $0.007 \pm 0.001$ | $0.037 \pm 0.002$ | $0.000 \pm 0.000$ | $0.009 \pm 0.001$ | $0.027 \pm 0.001$ | $0.000 \pm 0.000$ |
| | V-InFoR | $0.001 \pm 0.002$ | $0.046 \pm 0.003$ | $0.000 \pm 0.000$ | $0.006 \pm 0.003$ | $0.040 \pm 0.003$ | $0.000 \pm 0.000$ |
| | HINT-G | $0.003 \pm 0.000$ | $0.043 \pm 0.000$ | $0.000 \pm 0.000$ | $0.009 \pm 0.000$ | $0.047 \pm 0.000$ | $0.000 \pm 0.000$ |
| | ORExplainer | $\mathbf{0.018 \pm 0.002}$ | $\mathbf{0.034 \pm 0.001}$ | $0.000 \pm 0.000$ | $\mathbf{0.023 \pm 0.002}$ | $\mathbf{0.022 \pm 0.002}$ | $0.000 \pm 0.000$ |
| with unseen label nodes | GNNExplainer | $0.005 \pm 0.001$ | $0.040 \pm 0.001$ | $0.141 \pm 0.006$ | $-0.003 \pm 0.002$ | $0.038 \pm 0.002$ | $0.026 \pm 0.003$ |
| | PGExplainer | $0.010 \pm 0.001$ | $0.031 \pm 0.001$ | $0.078 \pm 0.003$ | $0.009 \pm 0.007$ | $0.018 \pm 0.002$ | $0.007 \pm 0.001$ |
| | MixupExplainer | $0.010 \pm 0.001$ | $0.032 \pm 0.001$ | $0.079 \pm 0.003$ | $0.010 \pm 0.007$ | $0.018 \pm 0.002$ | $\mathbf{0.005 \pm 0.002}$ |
| | ProxyExplainer | $0.010 \pm 0.001$ | $0.033 \pm 0.001$ | $0.074 \pm 0.003$ | $0.009 \pm 0.007$ | $0.019 \pm 0.002$ | $0.008 \pm 0.003$ |
| | V-InFoR | $0.001 \pm 0.001$ | $0.039 \pm 0.001$ | $0.174 \pm 0.004$ | $0.005 \pm 0.002$ | $0.032 \pm 0.006$ | $0.033 \pm 0.005$ |
| | HINT-G | $-0.002 \pm 0.000$ | $0.059 \pm 0.000$ | $0.174 \pm 0.007$ | $0.005 \pm 0.002$ | $0.049 \pm 0.000$ | $0.008 \pm 0.000$ |
| | ORExplainer | $\mathbf{0.020 \pm 0.000}$ | $\mathbf{0.029 \pm 0.000}$ | $\mathbf{0.062 \pm 0.000}$ | $\mathbf{0.026 \pm 0.000}$ | $\mathbf{0.016 \pm 0.002}$ | $0.007 \pm 0.001$ |

Table 11: Runtime (in seconds) reported as mean $\pm$ standard deviation.

| Method | BA-Community | Citeseer |
|---|---|---|
| GNNExplainer | $1.676 \pm 0.057$ | $1.028 \pm 0.032$ |
| PGExplainer | $0.319 \pm 0.025$ | $0.247 \pm 0.020$ |
| MixupExplainer | $0.538 \pm 0.013$ | $0.409 \pm 0.032$ |
| ProxyExplainer | $4.595 \pm 0.038$ | $2.711 \pm 0.118$ |
| V-InFoR | $0.513 \pm 0.025$ | $0.436 \pm 0.025$ |
| HINT-G | $50.209 \pm 2.686$ | $1.240 \pm 0.001$ |
| ORExplainer | $0.360 \pm 0.005$ | $0.241 \pm 2.591$ |

all methods, PGExplainer shows the shortest training time due to its simple architecture. Mixup-Explainer, ProxyExplainer, and V-InFoR incur additional overhead from data augmentation or the use of VGAE, while HINT-G is significantly slower because it requires influence score calculation for each node. ORExplainer requires slightly more time than PGExplainer but remains faster than the other baselines, demonstrating that the proposed *WEP* framework provides a clear runtime advantage.

# E   LIMITATIONS

While ORExplainer demonstrates strong robustness across diverse OOD scenarios, several limitations remain. First, since the proposed method relies on the energy scores derived from a pre-trained GNN, its effectiveness is inherently bounded by the reliability of the underlying model. When the pre-trained GNN suffers from severe distribution shifts that degrade its predictive performance, even ID nodes may be mischaracterized, making WEP less effective. Second, our evaluation has been limited to synthetic benchmarks and citation-style datasets; extending the analysis to more complex graph settings, such as dynamic or heterogeneous graphs, is an important direction for future work.

