# OpenReview forum: "Out-of-Distribution Robust Explainer for Graph Neural Networks"
_ICLR.cc/2026/Conference — Submitted to ICLR 2026_

### Official Review · Reviewer_1j5A · 2025-10-20

**Soundness:** 3
**Presentation:** 2
**Contribution:** 3
**Rating:** 4
**Confidence:** 4

**Summary:**

This paper proposes an OOD robust explainer for GNNs and introduces a WEP mechanism that suppresses unreliable OOD nodes while emphasizing in-distribution information. The authors also provide a theoretical analysis connecting WEP to a diffusion process. Experiments on synthetic and citation datasets show improved explanation stability and robustness compared to existing explainers.

**Strengths:**

1. The presentation of the paper is clear.

2. In the theoretical section, they try to connect the proposed mechanism with diffusion-based energy minimization, showing some effort toward providing interpretability and analytical grounding.

**Weaknesses:**

1. While the paper identifies three types of OOD nodes in Introduction, their definitions appear somewhat overlapping, as all are described in the context of new or injected nodes. It is unclear whether these categories are mutually exclusive or represent different perspectives of the same phenomenon. Clarifying the conceptual distinctions would improve the overall clarity of the problem setup.

2. The authors mention that existing explainers assume a fixed graph, but the failure modes of prior works are not clearly articulated. A deeper analysis of why such assumptions reduce robustness, and how the proposed method directly addresses these issues, would make the motivation more convincing.

3. Eq(3) defines WEP as a simple average between a node’s own energy and its neighbors’ energies. However, authors did not explain why this linear diffusion operation enhances robustness, or how many propagation steps k are used.

4. The theoretical derivation assumes that ID and OOD nodes have disjoint energy intervals. This assumption seems strong, and it is unclear whether it holds for real-world graphs where ID and OOD distributions may overlap.

5. Theorem 5.2 sets WEP as a lazy substochastic diffusion, showing that minimizing propagated energy reduces visits to high-energy nodes. However, this finding largely restates the intuitive effect of averaging over neighbors and does not offer a deeper theoretical guarantee of robustness or faithfulness. This analysis reads more like a mathematical restatement of the mechanism than a theory.

6. The experiments are conducted on small-scale datasets, which may not adequately test scalability or robustness on larger and more complex graphs. Including larger datasets would make the evaluation more convincing.

**Questions:**

Please refer to Weaknesses part.

---

> ### Author Response · Authors · 2025-11-21
>
> Thanks for your insightful comments and review.
>
>
> ## Weakness:
>
> > **W1:** While the paper identifies three types of OOD nodes in Introduction, their definitions appear somewhat overlapping, as all are described in the context of new or injected nodes. It is unclear whether these categories are mutually exclusive or represent different perspectives of the same phenomenon. Clarifying the conceptual distinctions would improve the overall clarity of the problem setup.
> >
>
> Graph data inherently consists of two complementary sources of information: structural connectivity and node features. For this reason, OOD behavior can arise along either dimension independently. Structural OOD corresponds to nodes whose connectivity patterns deviate significantly from the training distribution, while featural OOD captures cases where the features alone fall outside the learned distribution. Unseen-label OOD represents a different type of shift, where nodes belong to semantic classes not observed during training; such nodes can simultaneously exhibit both structural and featural deviations. Our categorization therefore reflects three conceptually distinct ways in which a node may diverge from the training distribution, without implying that these types are mutually exclusive. We will clarify this distinction in the revised version.
>
> > **W2:** The authors mention that existing explainers assume a fixed graph, but the failure modes of prior works are not clearly articulated. A deeper analysis of why such assumptions reduce robustness, and how the proposed method directly addresses these issues, would make the motivation more convincing.
> >
>
> When explaining graphs that contain OOD nodes, existing explainers may face two types of difficulties. The first arises from changes in the structural context. When structural OOD nodes are added, the target node can become connected within a few hops to nodes that belong to other motifs. In the original training distribution, such cross motif proximity does not occur, and the explainer only needs to reason over the structure inside the target’s own motif. With structural OOD present, the explainer must now consider a much larger and more complex set of possible subgraph combinations because nodes from unrelated motifs enter the target node’s receptive field. This expansion of the search space introduces patterns that the explainer has not encountered during training and increases the difficulty of identifying the correct explanatory subgraph. This tendency can be seen in Table 8 (page 18), where some baseline methods show noticeable declines in AUC even when their OOD precision remains low.
> The second type of difficulty appears in the representation space. Although the introduction of OOD nodes results in only small changes in the GNN’s overall prediction accuracy, as shown in Table 5 (page 17), their presence can still influence how in-distribution nodes are embedded. As shown in Table A, B, and C, the Silhouette Score of in-distribution embeddings becomes lower once OOD nodes are added, suggesting that class clusters that were previously well separated become less distinct. Such shifts can affect explainer models that rely heavily on node embeddings as input. In contrast, ORExplainer makes use of both the raw node features and the intermediate representations produced by the GNN, which provides a broader set of signals for the explainer to rely on and helps reduce its sensitivity to small representation shifts caused by OOD nodes. In addition, the WEP based energy loss encourages the explainer to place more weight on nodes whose representations align well with what the GNN can reliably make use of, which supports stable explanations even when parts of the representation space are perturbed.
>
> **Table A:** Silhouette score of embeddings on the synthetic datasets across different structure-level OOD
>
> | Structure-level OOD | BA-Shapes | BA-Comm | Tree-Cycle | Tree-Grid |
> |---------------------|-----------|---------|------------|-----------|
> | 0                   | 0.6810    | 0.4307  | 0.5161     | 0.6182    |
> | 10                  | 0.5925    | 0.4203  | 0.4881     | 0.5664    |
> | 20                  | 0.5528    | 0.4083  | 0.4549     | 0.5063    |
> | 30                  | 0.5219    | 0.3992  | 0.4375     | 0.4501    |
>
> **Table B:** Silhouette score of embeddings on the real world datasets across different feature-level OOD
> | Feature-level OOD | Cora   | Citeseer |
> |-------------------|--------|----------|
> | 0%                | 0.4246 | 0.3551   |
> | 10%               | 0.4189 | 0.3518   |
> | 20%               | 0.4145 | 0.3432   |
> | 30%               | 0.4135 | 0.3368   |
>
> **Table C:** Silhouette score of embeddings on the real world datasets across presence of unseen-label OOD nodes
> | Unseen-label OOD | Cora | Citeseer |
> |------------------|-----------|---------|
> | without          | 0.6054    | 0.5112  |
> | with             | 0.5960    | 0.5054  |

---

> > ### Author Response · Authors · 2025-11-21
> >
> > > **W3:** Authors did not explain why this linear diffusion operation enhances robustness, or how many propagation steps k are used.
> > >
> >
> > Eq. (3) defines WEP as a one-hop linear diffusion of energy scores. We use k propagation steps equal to the number of GNN layers, since node classification model aggregates information within k-hop neighborhoods, and the explanatory subgraph should reflect the same receptive field that influences the GNN’s prediction.
> > Energy scores are negative-valued, and lower energy indicates that a node is closer to the in-distribution region. When WEP is propagated over the k-hop neighborhood, the size regularizer constrains the total edge weight. Under this constraint, minimizing the target node’s propagated energy naturally encourages the explainer to assign larger weights to neighbors with low (more negative) energy and to suppress edges connected to high-energy neighbors. This behavior persists across multiple propagation steps.
> > Consequently, the diffusion mechanism causes ID nodes to dominate the explanatory subgraph while reducing the influence of OOD neighbors, which enhances robustness against OOD contamination.
> >
> > > **W4:** The theoretical derivation assumes that ID and OOD nodes have disjoint energy intervals. This assumption seems strong, and it is unclear whether it holds for real-world graphs where ID and OOD distributions may overlap.
> > >
> >
> > The assumption of disjoint energy intervals is introduced as a sufficient condition that simplifies the theoretical analysis, rather than as a requirement for the method to work. In practice, we do not expect perfect separation between ID and OOD energies, and indeed real-world datasets naturally exhibit overlap. The purpose of the assumption is to make the optimization objective analytically tractable and to illustrate the direction in which WEP pushes the explanation weights.
> > The method itself does not rely on strict interval separation. WEP operates by comparing relative energy levels within the local neighborhood and by suppressing higher-energy nodes through its propagation and regularization terms. This mechanism functions as intended even when the distributions overlap. Our empirical results confirm this behavior, showing that ORExplainer consistently down-weights OOD nodes despite imperfect energy separation.
> >
> >
> > > **W5:** Theorem 5.2 sets WEP as a lazy substochastic diffusion, showing that minimizing propagated energy reduces visits to high-energy nodes. However, this finding largely restates the intuitive effect of averaging over neighbors and does not offer a deeper theoretical guarantee of robustness or faithfulness. This analysis reads more like a mathematical restatement of the mechanism than a theory.
> > >
> >
> > We thank the reviewer for pointing this out. Our intention in Sec. 5 is not to claim a strong worst‑case robustness guarantee, but to formally clarify what the WEP regularizer is doing in terms of OOD exposure. Theorem 5.2 goes beyond a qualitative “averaging over neighbors” intuition by characterizing WEP as a lazy substochastic diffusion and explicitly linking the propagated energy at the target node to a path-based notion of OOD visitation. Concretely, under the standard energy-gap assumption used in energy-based OOD detection, Theorem 5.2 yields the inequality
> > $$
> > a\_{\mathrm{ID}} s\_t^{(k)} + \delta \phi\_{\mathrm{OOD}}^{(k)}(t) \le e\_t^{(k)} \le b\_{\mathrm{ID}} s\_t^{(k)} + (b\_{\mathrm{OOD}}-b\_{\mathrm{ID}}) \phi\_{\mathrm{OOD}}^{(k)}(t),
> > $$
> > where $\phi\_{\mathrm{OOD}}^{(k)}(t)$ is the total probability mass of $k$-step walks from the target $t$ that visit any OOD node. As we will clarify in the revision, this directly implies the corollary
> > $$
> > \phi\_{\mathrm{OOD}}^{(k)}(t) \le \frac{e\_t^{(k)} - a\_{\mathrm{ID}} s\_t^{(k)}}{\delta},
> > $$
> > i.e., minimizing the WEP energy term $\mathcal{L}\_{\mathrm{ene}} = e\_t^{(k)}$ provably upper-bounds the OOD visitation of diffusion paths from the target. This provides a formal justification that our robustness regularizer is a surrogate for suppressing OOD exposure in the explanation subgraph, which is empirically measured by the OOD edge precision metric in Sec. 7.
> >
> > In addition, we will make explicit that the cross-entropy term $\mathcal{L}\_{\text{CE}} = \mathrm{CE}(f(\mathcal{G}_t^\*), f(\mathcal{G}\_{\text{explain}}))$ is equivalent to minimizing $\mathrm{KL}(f(\mathcal{G}\_t^\*)) || f(\mathcal{G}\_{\text{explain}})$, thereby providing faithfulness in the sense that the predictive distribution on the target node is preserved up to small KL divergence. We will revise the Appendix to better highlight these links and to tone down any overstatement of “theoretical guarantees”; our main contribution remains empirical, while the analysis is meant as a formal design justification for WEP rather than a full robustness theory.

---

> ### Author Response · Authors · 2025-11-21
>
> > **W6:** The experiments are conducted on small-scale datasets, which may not adequately test scalability or robustness on larger and more complex graphs. Including larger datasets would make the evaluation more convincing.
> >
>
> To address the reviewer’s concern regarding scalability, we additionally conducted experiments on a larger real-world dataset, Amazon Photo. The results are reported in Table D. Following the same feature-level OOD configuration used in the main paper, we set the OOD ratio to 30 percent. For explanation extraction, we also follow the original experimental protocol and select the top 10 percent of edges as the explanation.
>
> Similar to the observations on Cora and Citeseer, ORExplainer achieved high explanation performance while including only a small number of OOD components in its explanations. Scalability with respect to graph size is unlikely to be an issue, because ORExplainer trains an explanation MLP that takes node features and embeddings as input.
> We will continue to strengthen the experimental evaluation by incorporating additional baselines and settings.
>
> **Table D:** The results on Amazon Photo under 30% feature-level OOD
> | Explainer      | $Fid\_{+}~(\uparrow)$ | $Fid\_{-}~(\downarrow)$ | $OOD~(\downarrow)$ |
> |----------------|-----------------------|-------------------------|--------------------|
> | GNNExplainer   | 0.032 ± 0.003         | -0.110 ± 0.013          | 0.485 ± 0.001      |
> | PGExplainer    | 0.026 ± 0.026         | -0.093 ± 0.047          | 0.507 ± 0.044      |
> | MixupExplainer | 0.030 ± 0.026         | -0.101 ± 0.046          | 0.504 ± 0.043      |
> | ORExplainer    | 0.078 ± 0.003         | -0.258 ± 0.002          | 0.000 ± 0.000      |

---

### Official Review · Reviewer_wAR2 · 2025-10-31

**Soundness:** 2
**Presentation:** 3
**Contribution:** 3
**Rating:** 4
**Confidence:** 3

**Summary:**

This paper proposes ORExplainer, a post-hoc explanation model designed to generate robust and reliable explanations for GNNs in the presence of out-of-distribution (OOD) nodes. The method introduces an Energy-Score mechanism to prioritize in-distribution (ID) nodes while suppressing OOD influence. The paper evaluates the approach on both synthetic and real-world datasets and reports improved robustness of explanations under several OOD settings. The topic is timely and relevant to trustworthy graph explainability. However, the paper contains several writing, methodological, and conceptual issues that need to be clarified before the contribution can be properly assessed.

**Strengths:**

1. Addresses an important and underexplored problem—robust explainability under graph OOD conditions.
2. The use of Energy Scores for node importance is intuitively reasonable.
3. The experiments include both synthetic and real-world datasets, demonstrating practical relevance.

**Weaknesses:**

1. Writing clarity issues.
Some expressions are grammatically or semantically unclear. For example: Line 126: should read “The graph used for explanation …” instead of the current phrasing. Line 129: should use “The model f is a node classifier …” rather than “The GNN f is a …”.

2. Unclear generation of OOD settings.
Lines 301–307 only describe how synthetic node OOD and real-world feature OOD are generated. It remains unclear:
How are node OODs created in real datasets?
How are feature OODs constructed in synthetic datasets?
For synthetic “unseen-label” OOD, is the largest label simply treated as unseen?

Furthermore, the experimental design is inconsistent: synthetic datasets only test structural OOD, while real datasets only test feature and unseen-label OOD. Why not evaluate all three OOD types on both domains to support the claim of general robustness?

3. Questionable assumption on excluding OOD nodes.
 Lines 199–202 state that explanations should consist mainly of ID nodes, with OOD nodes excluded or down-weighted. However, this assumption may fail when OOD nodes directly cause prediction errors. In such cases, removing them may hide the model’s failure mechanism rather than provide a faithful explanation. The authors should discuss this limitation explicitly.

4. Missing extension discussion.
Could the proposed ORExplainer framework be extended to graph-level classification tasks? Since the method currently focuses on node-level explanations, a discussion of potential extensions would be valuable.

5. Target-node OOD scenario.
If the target node itself is OOD, can ORExplainer still produce a reliable and meaningful explanation? This scenario seems practically important, but is not analyzed in the paper.

**Questions:**

See Weaknesses.

---

> ### Author Response · Authors · 2025-11-21
>
> Thanks for your insightful comments and review.
>
> ## Weakness:
>
> > **W1:** Writing clarity issues.
> >
>
> We have corrected the expressions pointed out by the reviewer and revised the manuscript accordingly. The revised parts are highlighted in red in the paper. We will continue to refine the writing throughout the paper to improve clarity and readability.
>
> > **W2:** Unclear generation of OOD settings.
> >
>
> To ensure that each dataset was evaluated under an OOD setting appropriate to its characteristics, we applied feature-level and unseen-label OOD settings to the real-world datasets, and structure-level OOD to the synthetic datasets. We address the reviewer’s detailed questions about each OOD construction below.
>
> > **W2-1:** How are node OODs created in real datasets?
> >
>
> In our experimental settings, node-level OOD in real-world datasets is not created by adding new nodes. Instead, we select a subset of existing nodes and assign OOD properties to them.
> For feature-level OOD, we replace the original feature values of selected nodes with high-intensity noise. The noise level is set to be roughly twice the average feature intensity. For example, in Cora, the average number of active features is about 18. We therefore assign the value 1 to 36 randomly chosen feature dimensions when constructing OOD nodes.
> For unseen-label OOD, we mark all nodes belonging to the largest label as OOD. This is done without any additional modification to their features or connectivity.
> To ensure that these nodes represent distributions unseen during node classifier training, we remove the selected OOD nodes and their incident edges when training the node classifier $f$. During explainer training and evaluation, we restore the removed nodes and edges so that the explainer encounters OOD nodes.
>
> > **w2-2:** How are feature-level OODs constructed in synthetic datasets?
> >
>
> Feature-level OOD is not constructed in synthetic datasets because these benchmarks use simple, low-dimensional indicator features and are designed specifically to evaluate structure-based reasoning. Since the primary signal in synthetic graphs comes from structural motifs, defining a meaningful feature-level OOD shift in this setting is not appropriate.
>
> > **W2-3:** For synthetic “unseen-label” OOD, is the largest label simply treated as unseen?
> >
>
> Unseen-label OOD is also not applicable to synthetic datasets. The node labels correspond to structural roles—motif nodes or non-motif background nodes—rather than semantic classes used by the node classifier. Removing any motif label would break the motif itself, since motifs require all role-specific nodes to form a coherent pattern. Removing the non-motif label would eliminate the background structure and leave only isolated motifs, producing a degenerate graph that leads to trivial overfitting. In both cases, hiding a label would distort the intended structural distribution rather than creating a meaningful unseen-label split. For this reason, structure-level OOD is the only valid OOD type for synthetic datasets.
>
> > **W3:** The experimental design is inconsistent. synthetic datasets only test structure-level OOD, while real datasets only test feature and unseen-label OOD. Why not evaluate all three OOD types on both domains to support the claim of general robustness?
> >
>
> As discussed above, not all OOD types are meaningfully defined across both synthetic and real-world datasets. Synthetic benchmarks are designed to evaluate structure-driven reasoning and therefore naturally support only structure-level OOD. Feature-level or unseen-label OOD cannot be constructed in a meaningful way without breaking the motif-based generation rules.
> Real-world datasets such as Cora and Citeseer, however, exhibit different characteristics. Sometimes their predictive signal largely comes from high-dimensional feature vectors rather than fine-grained structural patterns. [1]
> In conclusion, generating structure-level OOD cases in real-world datasets that are disentangled from node features is challenging, primarily because the rich node features tend to overshadow or compensate for structural variations. This is due to the advent of abnormal node features induced by atypical neighboring nodes, which may also alter the original prediction of the target GNN.
> For these reasons, we evaluate OOD types that are appropriate for each domain. Structure-level OOD is the valid and informative setting for synthetic datasets, while feature-level and unseen-label OOD are the meaningful forms of distribution shift for real citation graphs. Our experimental design therefore reflects the characteristics of each dataset rather than omitting OOD scenarios arbitrarily.

---

> ### Author Response · Authors · 2025-11-21
>
> > **W4:** Questionable assumption on excluding OOD nodes. Lines 199–202 state that explanations should consist mainly of ID nodes, with OOD nodes excluded or down-weighted. However, this assumption may fail when OOD nodes directly cause prediction errors. In such cases, removing them may hide the model’s failure mechanism rather than provide a faithful explanation. The authors should discuss this limitation explicitly.
> >
>
> We focus on explanations for cases where the node classifier still makes a correct and stable prediction even when OOD nodes exist around the target. If the prediction does not change, it indicates that the GNN is not actually using those OOD nodes, and including them in the explanation would be misleading. As stated in our experimental setup (Line 863), we therefore generate explanations only for nodes whose predictions remain correct after OOD contamination.
> If OOD nodes do change the prediction, this represents a fundamentally different type of problem. The node classifier is no longer behaving reliably, and the OOD nodes become part of a failure case rather than part of a normal decision process. Explaining such failure-inducing OOD nodes is valuable, but it corresponds to analyzing model breakdown under severe distribution shift rather than producing faithful explanations for correct predictions. This lies outside the scope of our work, which focuses on stable explanations under preserved predictions.
> We will clarify this distinction in the revised paper.
>
> > **W5:** Missing extension discussion. Could the proposed ORExplainer framework be extended to graph-level classification tasks? Since the method currently focuses on node-level explanations, a discussion of potential extensions would be valuable.
> >
>
> The current framework is designed for node-level explanations because the OOD score is computed at the node level, and the energy-based formulation depends on node-wise logits. For this reason, the proposed method cannot be directly applied to graph-level classification tasks.
> However, the underlying idea—suppressing out-of-distribution information when generating explanations—is not restricted to the node-level setting. A graph-level extension would be feasible by constructing distribution-based scores over node sets or subgraphs (e.g., aggregating node-wise energy signals or defining subgraph-level OOD measures). We believe that incorporating such aggregated OOD scoring mechanisms would allow ORExplainer to be extended naturally to graph-level explanations.
>
> > **W6:** Target-node OOD scenario. If the target node itself is OOD, can ORExplainer still produce a reliable and meaningful explanation? This scenario seems practically important, but is not analyzed in the paper.
> >
>
> If the target node itself is OOD, the node classifier is no longer operating within the distribution it was trained to handle, and its prediction becomes inherently unreliable. In this situation, the model's output cannot be judged as correct or incorrect, because an OOD target does not have a well-defined ground-truth label with respect to the training distribution. As a result, any explanation generated for such a prediction cannot be regarded as stable or meaningful.
> In practice, explaining an OOD target node corresponds to analyzing a failure case of the model under severe distribution shift rather than explaining its intended decision-making process. This is fundamentally different from the goal of our work. Consistent with our experimental protocol (Line 863), we generate explanations only for nodes whose predictions remain valid after OOD nodes are added, ensuring that the explanation problem itself is well-defined.
>
> [1] Graph Neural Networks Use Graphs When They Shouldn't, ICML2025

---

### Official Review · Reviewer_Fv2n · 2025-10-31

**Soundness:** 3
**Presentation:** 3
**Contribution:** 3
**Rating:** 6
**Confidence:** 3

**Summary:**

This paper studied the problem of generating robust post-hoc, instance-level explanations for Graph Neural Networks (GNNs) in dynamic settings，where new nodes/edges at test time can introduce out-of-distribution (OOD) noise and outliers, undermining existing XAI methods that assume distributional consistency. To address this challenge, the authors propose ORExplainer, which incorporates Energy Scores to capture structural dependencies, prioritize in-distribution nodes, and suppress the influence of OOD nodes during explanation generation. Experiments across synthetic and real-world datasets with varied OOD injection strategies show that ORExplainer delivers more reliable and robust explanations than prior approaches, and the implementation is released for reproducibility.

**Strengths:**

1. This paper proposes ORExplainer, which provides robust and verifiable instance level explanations for GNNs under test time OOD scenarios in dynamic graphs. The problem setting is novel, and the method is supported by solid experimental results and theoretical analysis.

2. The paper is generally readable and reasonably well structured.

**Weaknesses:**

1. Some experimental results appear to be insufficiently comprehensive. For example, Figure 3 shows only the BA-Community and Cora datasets.

2. The paper appears not to report the accuracy metrics commonly used by prior GNNExplainer.

**Questions:**

1. Some experimental results appear insufficiently comprehensive. For example, Figure 3 includes only BA-Community and Cora; please expand to additional datasets or justify this selection.

2. The paper does not report the accuracy metrics commonly used by prior explainers (e.g., GNNExplainer). Please justify this choice and, if appropriate, include those metrics for comparison.

3. In Table 1 and Table 3, some results appear less than ideal. Please explain the underlying reasons.

---

> ### Author Response · Authors · 2025-11-21
>
> Thanks for your insightful comments and review.
>
> ## Questions:
>
> > **Q1:** Some experimental results appear insufficiently comprehensive.
> >
>
> The complete experimental results for BA-Community, Cora, and all additional OOD settings and datasets are reported in Tables 8–10 in the Appendix D (pages 18–19). Due to space constraints in the main paper, Figure 3 includes only two representative datasets. The extended results show trends consistent with those in the main text: across a wide range of datasets and OOD conditions, ORExplainer consistently demonstrates higher faithfulness and robustness compared to existing baselines. We hope the supplementary results provide a clearer picture of the overall generality of our method.
> We will explicitly mention that the additional experiments are provided in Appendix D (pages 18-19).
>
> > **Q2:** Please justify choice of evaluation metrics.
> >
>
> The accuracy metric reported in GNNExplainer evaluates whether each predicted edge matches the ground-truth (GT) explanatory subgraph. In the synthetic setting, this becomes a binary classification task (GT edge vs. non-GT edge), and usually the number of non-GT edges outweighs the GT edges. Since AUC is generally more stable and informative than raw accuracy, especially under strong class imbalance as in the synthetic settings, PGExplainer [1] and subsequent works [2–5] adopt AUC as the standard metric. For this reason, we follow the established convention and report AUC on synthetic datasets.
> For real-world datasets, ground-truth explanations are not available, making accuracy-based metrics inapplicable. In such cases, Fidelity has been widely used in prior XAI literature to quantify explanation quality by measuring changes in model predictions. We therefore adopt Fidelity for real-world experiments in accordance with standard practice.
>
> > **Q3:** In Table 1 and Table 3, some results appear less than ideal. Please explain the underlying reasons.
> >
>
> When we consider the results in Tables 8 through 10 in the appendix, the overall trend becomes clear. As the degree or proportion of OOD nodes increases, ORExplainer consistently achieves lower OOD-edge precision and higher explanation quality than the baselines. This indicates that ORExplainer remains comparatively more robust as OOD contamination becomes stronger and supports its effectiveness under distribution shift.
> Some entries in Table 1 and Table 3 show slightly lower values, but these differences arise primarily from characteristics of the datasets. In the unseen-label setting on Citeseer, node labels correspond to research domains with overlapping semantic and feature patterns. Because of this overlap, it is inherently difficult for the energy score to cleanly separate unseen-label nodes from in-distribution nodes. This results in very small fluctuations, such as an OOD-edge precision difference of about 0.002, which we view as expected and not indicative of a significant performance issue.
> In structure-level OOD settings, the injected nodes not only introduce new edges directly connected to the target node but also expose additional connectivity patterns that were not present during training. These multi-hop structural changes alter the surrounding motif relationships and cause baseline explainers to become highly sensitive to these newly formed connections. As a result, their AUC drops substantially once structure-level OOD is added. ORExplainer is less affected by these indirect connectivity changes because its energy-based mechanism suppresses the influence of OOD-related message passing, allowing its explanation quality to remain more stable even when the structural environment around the target becomes more complex. Although the OOD-edge precision in this setting appears close across methods (baseline: 0.006 ± 0.002, ours: 0.007 ± 0.000), the absolute values are small, and the difference becomes even less meaningful once the variance is taken into account. In contrast, the AUC shows a large and consistent improvement for ORExplainer, indicating that its overall explanation quality is substantially more robust to structural perturbations than that of the baselines.
>
> [1] Parameterized Explainer for Graph Neural Network, NeulPS2020
>
> [2] MixupExplainer: Generalizing Explanations for Graph Neural Networks with Data Augmentation, SIGKDD2023
>
> [3] Generating In‑Distribution Proxy Graphs for Explaining Graph Neural Networks (ProxyExplainer), ICML 2024
>
> [4] HINT‑G: Harnessing Influence Function for Task‑Irrelevant Explanation on Graph Neural Networks, KDD2-25
>
> [5] XgExplainer: Robust Evaluation‑based Explanation for Graph Neural Networks, SDM2024

---

> > ### Comment · Reviewer_Fv2n · 2025-11-27
> > **Response to the Authors**
> >
> > Thanks for the clarifications. I prefer to maintain my initial evaluations at this stage.

---

### Official Review · Reviewer_RZhh · 2025-11-01

**Soundness:** 2
**Presentation:** 2
**Contribution:** 2
**Rating:** 4
**Confidence:** 4

**Summary:**

This paper proposes ORExplainer, an out-of-distribution (OOD) robust post-hoc explainer for graph neural networks (GNNs). It introduces Weighted Energy Propagation (WEP) based on node energy scores to suppress unreliable OOD nodes and enhance explanation reliability. The method provides stable, in-distribution–focused subgraph explanations even under noisy or OOD conditions. Extensive experiments on synthetic and real-world datasets demonstrate that ORExplainer outperforms existing explainers in both robustness and fidelity.

**Strengths:**

1. The proposed Weighted Energy Propagation (WEP) effectively suppresses OOD influence, offering a simple yet principled way to enhance explainer robustness.

2. Extensive experiments on diverse datasets show consistent improvements in both robustness and fidelity over existing methods.

**Weaknesses:**

1. The overall writing quality could be improved. For example: (1) the caption of Figure 2 is not expressed clearly; (2) the font style of the embedding matrix Z in Preliminaries should be unified; (3) abbreviations such as “out-of-distribution (OOD)” appear repeatedly across Introduction, Related Work, and Our Proposed Method; and (4) subsection formatting should be made consistent.

2. In Figure 2, the CE loss should also have an arrow pointing to ORExplainer, and the main diagram could benefit from more detailed and polished visual design.

3. The experimental section could be strengthened by adding quantitative analyses of the energy mechanism to demonstrate its contribution and effectiveness.

**Questions:**

1.The method ensures robustness by suppressing OOD message passing, but such OOD information may sometimes carry useful signals. It would be valuable to discuss how these potentially informative OOD components could be better utilized.

---

> ### Author Response · Authors · 2025-11-21
>
> Thanks for your insightful comments and review.
>
> ## Weakness:
>
> > **W1:** The overall writing quality could be improved.
> >
>
> We appreciate the reviewer’s feedback regarding writing clarity. We have revised the indicated parts accordingly and marked the updates in red in the paper. We will continue to refine the wording throughout the paper to improve overall readability and presentation quality.
>
>
> > **W2:** In Figure 2, the CE loss should also have an arrow pointing to ORExplainer, and the main diagram could benefit from more detailed and polished visual design.
> >
>
> The CE-loss connection has been added to Figure 2, and we have refined the overall layout to improve visual clarity and polish.
>
>
> > **W3:** The experimental section could be strengthened by adding quantitative analyses of the energy mechanism to demonstrate its contribution and effectiveness.
> >
>
> We conducted ablation studies to quantify the contribution of WEP.
> Here, $\mathcal{L}\_{\mathrm{reg}}$ denotes the combination of $\mathcal{L}\_{\mathrm{size}}$ and $\mathcal{L}\_{\mathrm{ent}}$, which prevents the trivial solution where all edge weights collapse to 1; therefore, it is included in all settings.
>
> Across all datasets and OOD types, the model trained with both $\mathcal{L}\_{\mathrm{CE}}$ and $\mathcal{L}\_{\mathrm{ene}}$ consistently achieves the lowest OOD-edge precision. This shows that combining prediction supervision with WEP is essential for removing OOD nodes while maintaining explanation quality.
>
> Table A shows that on BA-Community (structure-level OOD), adding WEP substantially improves both AUC and OOD-edge precision compared to using $\mathcal{L}\_{\mathrm{CE}}$ alone. The full loss combination achieves the strongest performance (AUC 0.982 and the lowest OOD precision), indicating that WEP effectively suppresses structurally inconsistent nodes introduced by OOD injection.
>
> As shown in Table B, in the feature-level OOD condition, using only $\mathcal{L}\_{\mathrm{ene}}$ fails to maintain fidelity and increases OOD selection, whereas the full loss combination restores fidelity and yields the lowest OOD precision. A similar pattern appears in the unseen-label OOD setting, as shown in Table C. WEP alone is insufficient, but becomes effective when combined with $\mathcal{L}\_{\mathrm{CE}}$.
>
> Overall, the ablations indicate that WEP contributes meaningfully to reducing OOD edges, and its benefit becomes more evident when combined with $\mathcal{L}\_{\mathrm{CE}}$ and the regularizers. Using the full objective leads to consistently stable explanation performance across the evaluated datasets.
>
> ## Questions:
>
> > **Q1:** The method ensures robustness by suppressing OOD message passing, but such OOD information may sometimes carry useful signals. It would be valuable to discuss how these potentially informative OOD components could be better utilized.
> >
>
> Our work focuses on generating stable and robust explanations when the graph contains OOD components, while the GNN's prediction remains unchanged even when additional OOD nodes are introduced. In this condition, the unchanged prediction suggests that the OOD components have a limited effect on the model's decision-making process. Therefore, we argue that including such OOD nodes in the explanation reduces faithfulness, which matches our experimental setting.
>
> We acknowledge that OOD components may carry useful signals in cases where their addition changes the prediction. In such scenarios, those OOD nodes become part of the actual decision and should indeed be highlighted by an explainer. However, analyzing situations where OOD nodes alter the model's prediction corresponds to understanding the target GNN model's failures under severe distribution shift. This is a distinct research direction from our focus on maintaining faithful explanations of the explainer model when the prediction remains stable under moderate OOD contamination. We will clarify this distinction in the revision.

---

> ### Author Response · Authors · 2025-11-21
>
> **Table A:** The results of ablation study on BA-Community with 30 structure-level OOD nodes.
> |                                     Loss Combination                                     | $AUC~(\uparrow)$ | $OOD~(\downarrow)$ |   |   |
> |:----------------------------------------------------------------------------------------:|:----------------:|:------------------:|---|---|
> | $\mathcal{L}\_{\mathrm{CE}} + \mathcal{L}\_{\mathrm{reg}}$                               | 0.663 ± 0.036    | 0.069 ± 0.010      |   |   |
> | $\mathcal{L}\_{\mathrm{ene}} + \mathcal{L}\_{\mathrm{reg}}$                              | 0.865 ± 0.025    | 0.039 ± 0.006      |   |   |
> | $\mathcal{L}\_{\mathrm{CE}} + \mathcal{L}\_{\mathrm{ene}} + \mathcal{L}\_{\mathrm{reg}}$ | 0.982 ± 0.003    | 0.019 ± 0.004      |   |   |
>
>
> **Table B:** The results of ablation study on Citeseer with 30% feature-level OOD nodes.
> |                                     Loss Combination                                     | $Fid\_{+}~(\uparrow)$ | $Fid\_{-}~(\downarrow)$ | $OOD~(\downarrow)$ |   |
> |:----------------------------------------------------------------------------------------:|:---------------------:|:-----------------------:|:------------------:|---|
> | $\mathcal{L}\_{\mathrm{CE}} + \mathcal{L}\_{\mathrm{reg}}$                               | 0.021 ± 0.000         | 0.007 ± 0.001           | 0.016 ± 0.002      |   |
> | $\mathcal{L}\_{\mathrm{ene}} + \mathcal{L}\_{\mathrm{reg}}$                              | 0.006 ± 0.000         | 0.013 ± 0.000           | 0.054 ± 0.002      |   |
> | $\mathcal{L}\_{\mathrm{CE}} + \mathcal{L}\_{\mathrm{ene}} + \mathcal{L}\_{\mathrm{reg}}$ | 0.019 ± 0.001         | 0.005 ± 0.001           | 0.007 ± 0.001      |   |
>
>
> **Table C:** The results of ablation study on Cora with unseen-label OOD nodes.
> |                                     Loss Combination                                     | $Fid\_{+}~(\uparrow)$ | $Fid\_{-}~(\downarrow)$ | $OOD~(\downarrow)$ |
> |:----------------------------------------------------------------------------------------:|:---------------------:|:-----------------------:|:------------------:|
> | $\mathcal{L}\_{\mathrm{CE}} + \mathcal{L}\_{\mathrm{reg}}$                               | 0.022 ± 0.001         | 0.026 ± 0.001           | 0.073 ± 0.001      |
> | $\mathcal{L}\_{\mathrm{ene}} + \mathcal{L}\_{\mathrm{reg}}$                              | 0.007 ± 0.001         | 0.020 ± 0.001           | 0.183 ± 0.006      |
> | $\mathcal{L}\_{\mathrm{CE}} + \mathcal{L}\_{\mathrm{ene}} + \mathcal{L}\_{\mathrm{reg}}$ | 0.020 ± 0.001         | 0.029 ± 0.001           | 0.062 ± 0.005      |

---

### Author Response · Authors · 2025-11-21

Please note that the revised parts of the paper are highlighted in red.

---

### Author Response · Authors · 2025-12-03
**Rubuttal summary of AC**

We are grateful for the thoughtful and constructive reviews provided by all reviewers. Their assessments highlight several strengths of our work as:
1. Addresses an important and unexplored problem: generating stable and faithful explanations under test-time OOD conditions.
2. Introduces WEP, a simple and principled mechanism for suppressing the impact of OOD components with energy-based grounding.
3. Demonstrates consistent gains in robustness and fidelity across synthetic and real-world datasets.

However, the reviewers also identified several vital points that require clarification:
> 1. Settings of OOD construction and generation.

Reviewer wAR2 raised questions about OOD experience settings. We clarified that our design intentionally reflects the characteristics of each dataset. Synthetic benchmarks focus on structural signals, so applying structure-level OOD is well-aligned. In contrast, real-world graphs depend on high-dimensional features, making feature-level and unseen-label OOD more appropriate. We also clarified that, for real-world datasets, OOD generation is performed by injecting noise into the features of existing nodes or by masking an entire label.
> 2. Treatment of OOD nodes in explanations.

Several reviewers (RZhh, wAR2) ask whether excluding OOD nodes from explanations is appropriate. As clarified in the revised manuscript, our problem setting focuses on explaining ID target nodes whose predictions remain stable even under OOD contamination. In this case, the stable prediction indicates that the OOD nodes did not have a noticeable effect on the GNN’s prediction, and including them in the explanation would be misleading.
Reviewer wAR2 also questioned the scenario where the target node itself is OOD. In such a case, the classifier is no longer operating within the distribution on which it was trained. The resulting prediction is unreliable, making any explanation derived from it difficult to regard as reliable. For this reason, we emphasized that our work focuses on explaining ID targets under OOD contamination rather than handling failure cases caused by OOD targets.
> 3. Evaluation metrics, scalability, and task extensions

Reviewer Fv2n  raised questions regarding the choice of evaluation metrics. For synthetic datasets with ground-truth edges, we use AUC, which is commonly used in prior work. For real-world datasets without ground truth, we use Fidelity, the standard metric for explanation faithfulness.
For the large-scale concern raised by Reviewer 1j5A, we evaluated ORExplainer on the Amazon Photo dataset under a 30% feature-level OOD setting. ORExplainer maintained strong explanation performance despite including only a small number of OOD edges, indicating that ORExplainer is scalable to graph size.
Reviewer wAR2 also asked about extending our method to graph-level classification. While the current formulation computes OOD scores at the node level and therefore cannot be applied directly, the underlying idea can be extended by aggregating node-wise energy measures or defining subgraph-level OOD scores. This provides a clear path toward adapting ORExplainer to graph-level tasks.
> 4. Failures of prior explainers under OOD Conditions

Reviewer 1j5A requested a clear articulation of the failure modes of prior work under OOD conditions, and we responded by explaining that existing explainers become unstable for two main reasons.
Structure-level OOD introduces unseen connectivity patterns, enlarging candidate subgraph space. Feature-level OOD also blurs the representation space: as shown in Table 5 and Table A (reviewer 1j5A).
ORExplainer mitigates these effects by leveraging raw features and intermediate embeddings, and by using the WEP loss to emphasize nodes whose representations remain reliable for the underlying GNN.
> 5. Effect of Weighted Energy Propagation

Reviewers (RZhh, 1j5A) asked for an explanation of WEP’s impact.
Our ablation studies show that WEP reduces OOD edges, and its contribution becomes more apparent when combined with $\mathcal{L}_{\mathrm{CE}}$ and the regularizers. Using the full objective consistently yields the most stable explanation performance across all evaluated datasets.
This behavior stems from how WEP propagates energy scores. Minimizing the propagated energy encourages the explainer to assign higher weights to low-energy neighbors while reducing high-energy ones, and the size regularizer constrains the total edge weight to avoid trivial solutions.
As a result, WEP naturally highlights ID nodes and suppresses the influence of OOD components, producing explanations that remain more reliable under OOD contamination.
> 6. Clarification of Theorems

Reviewer 1j5A also questioned the depth of the theoretical analysis, particularly the role of Theorem 5.2. Regarding Theorem 5.2, our aim is to formally clarify how the WEP regularizer works in terms of OOD exposure. We clarify these points in the revision and appropriately moderate the theoretical claims.

---

### Meta-Review · Area_Chair_n8UW · 2025-12-16

**Summary:**

(*Disclaimer: given the peculiar review process, some of my choices and reasonings below will be highly subjective, as I tried to imagine how a reviewer would have reacted to a specific response. I understand that any negative choice will be perceived as unfair by the authors, and I apologize in advance for that.*)

(*Second disclaimer: the authors and some reviewers explicitly mention some changes in scores that occurred during the rebuttal. As these were reverted due to the possibility of collusion in light of the security incident, I will tend to disregard this information.*)

The paper introduces a post-hoc explainer for graph neural networks (GNNs), in the scenario where some nodes and features may be OOD (during testing, but not during training of the explainer). This is obtained by designing a specific energy function whose minimization is directly linked with reducing the influence of OOD elements. The algorithm is tested on several benchmarks.

The initial reviews were clustered around a weak rejection. Among other things, the reviewers questioned the writing quality (as well as figure clarity), some unmotivated claims in the introduction, the evaluation in a small-scale setup, and in general the validity of the energy formulation.

The (partial) rebuttal period was inconclusive, since 3 out of 4 reviewers never engaged with the authors. This makes it hard to evaluate how their scores would have evolved in the final discussion period.

As I discuss below, I believe most of the issues raised by the reviewers are still valid. For example, the paper still has several typos (e.g., "these scenarios represent realistic challenges are useful", "explantions"), despite this being a concern of two reviewers. The authors discuss at length the need to handle node-level, feature-level, or label-level OOD, but the energy function combines all scenarios by defining a single scalar for each node (`1j5A`). In addition, the theoretical justification for the energy function makes an assumption on the energy levels that trivializes the result (`1j5A`).

Overall, even in the most optimistic scenario I do not believe the reviewers would have uniformly moved towards a positive evaluation of the paper.

**Reviewer Concerns:**

(*I will focus on some key weaknesses identified by multiple reviewers.*)

**Energy function** (`1j5A`, `RZhh`) : `1j5A` questioned the definition of the function in relation to the separate OOD scenarios described in the introduction; and the assumption on the different energy intervals in the theorem. `RZhh` asked for additional visualizations in the experimental evaluation. These points are valid and I do not believe the rebuttal addresses them in sufficient depth.

**Small-scale evaluation** (`1j5A`): the authors added a new dataset (Amazon Photos) in the rebuttal, but the evaluation is limited compared to the other datasets.

**Writing quality** (`wAR2`, `RZhh`): see above, this is still an issue.

**Experimental setup** (`wAR2`, `Fv2n`): the authors focus on a scenario where (a) OOD nodes are not present during training, and (b) OOD nodes are present during inference but are assumed not to influence the predictions (as they focus on correct predictions). This is a relatively narrow focus that was questioned by `wAR2`. `Fv2n` asked for additional metrics, and I think the answer was comprehensive.

**Reviewer Scores:**

`1j5A`: the reviewer did not engage in the rebuttal. However, given the concerns discussed above, I believe their recommendation would have remained towards a rejection.

`wAR2`: same as `1j5A`.

`Fv2n`: the reviewer had relatively small concerns that were addressed in the rebuttal.

`RZhh`: same as `Fv2n`.

---

### Decision · Program_Chairs · 2026-01-26

Reject